# INFLUENCE-AWARE FORECASTING: BREAKING THE SELF-STIMULATION BARRIER IN TIME SERIES

## ABSTRACT

The field of time series forecasting faces a critical performance plateau, where even billion-parameter foundation models struggle to outperform simple linear baselines. We argue this stagnation stems not from model architecture but from a universally adopted yet flawed 'self-stimulation' assumption, where models ignore the external influences that drive real-world systems by predicting the future using only the historical values of time series. Through a control-theoretic lens, we formally prove that this assumption imposes a hard, mathematical barrier on forecasting accuracy. To break this barrier, we introduce Influence-Aware Time Series Forecasting (IATSF), a new paradigm that reframes the task from correlation-based inference to dynamic system modeling. To operationalize this paradigm, we provide two foundational contributions. First, we introduce a leak-free, temporally-synced benchmark—a critical resource for the community—that incorporates textual influences to capture the qualitative or uncertain dynamics missed by traditional variables. Second, we develop FIATS, a lightweight, principled model engineered to interpret these influences. Its novel channel-aware mechanisms allow it to adjust its sensitivity to both textual signals and historical data in a channel-specific manner. Our results demonstrate that explicitly modeling external influences is not just an incremental improvement but the primary path forward for meaningful progress in time series forecasting.

## 1 INTRODUCTION

The field of time series forecasting has reached a critical performance plateau. Despite the development of sophisticated deep learning architectures (Nie et al., 2023; Liu et al., 2023; Jin et al., 2023) and even billion-parameter foundation models (Ansari et al., 2024; Shi et al., 2025; Woo et al., 2024), these advanced models deliver only marginal performance gains over simple linear baselines (Zeng et al., 2023; Xu et al., 2023; Toner & Darlow, 2024). We contend this lack of progress is not an issue of model complexity, but stems from a universally adopted yet flawed assumption: "self-stimulation," where models predict the future using only the historical time series obserevation, thereby ignoring the external influences that drive real-world systems.

Through a control-theoretic lens, we show that this assumption imposes a mathematical barrier on forecasting accuracy. By implicitly treating unobserved influences as random noise, traditional models are mathematically constrained to predict a blurry, "averaged-out" future, ignoring the sharp patterns caused by specific real-world events. While incorporating pre-defined exogenous variables is a step forward (Arango et al., 2025; Wang et al., 2024b), this approach often lacks the flexibility to capture the nuanced, non-quantifiable events that drive system dynamics. More recent work has turned to textual data (Williams et al., 2025; Aksu et al., 2024; Wang et al., 2024a), but these approaches, particularly those leveraging large language models (LLMs), have often lacked a rigorous theoretical grounding for how influences should be modeled. Our analytical framework provides this missing foundation, explicitly demonstrating that incorporating influence-related context is essential to lower the forecasting error bound.

To break this barrier, we introduce **Influence-Aware Time Series Forecasting (IATSF)**, a new paradigm that reframes the task from merely continuing observed patterns to modeling the dynamic system that generates them. We focus on textual data due to its ubiquity and its ability to encode nuanced, non-quantifiable signals often missed by traditional variables. This approach aligns forecasting with real-world system dynamics and unlocks new potential for interpretability and adaptability.

Despite these theoretical advances, practical adoption remains challenging due to the lack of datasets and models that are compatible with influence-aware forecasting. Existing multimodal time series forecasting (TSF) approaches often rely on large language models (LLMs) and datasets (Liu et al., 2024a; Williams et al., 2025) optimized for prompting rather than structured influence modeling. Consequently, these datasets often have: (1) short horizons limiting meaningful influence evaluation; (2) overly simplistic or ambiguous textual descriptions causing information leakage or irrelevance; and (3) poor temporal synchronization between textual and numerical data. To address these limitations and operationalize our theoretical insights, we introduce the Temporal-Synced IATSF benchmark, explicitly designed with leak-free textual influences synchronized to extended, realistic forecasting horizons.

To demonstrate the effectiveness of influence-aware forecasting, we propose *FIATS (Forecaster for Influence-Aware Time Series)*, a lightweight, *LLM-free* baseline model designed as an architectural embodiment of our control-theoretic principles. Inspired by our analysis, FIATS reframes the cross-attention mechanism to explicitly model the control process where external influences guide system dynamics. This is achieved through novel mechanisms, including a Channel-Aware Adaptive Sensitivity Modeling (CASM) mechanism and an Influence-Modulated Decoder with Channel-Aware Parameter Sharing (CAPS). This principled design enables FIATS to learn each channel's specific sensitivity to an influence and apply this insight to the forecast, directly operationalizing our theoretical findings.

Our extensive experiments across synthetic, physics-based, and market datasets demonstrate that modeling external influences is not just an incremental improvement but the primary path forward for meaningful progress in time series forecasting. Our key contributions are:

- A control-theoretic analysis that reveals intrinsic forecasting barriers caused by the "self-stimulation" assumption and proves that influence-aware modeling reduces error bounds.
- The introduction of IATSF, a paradigm that models time series with external influences, bridging the gap between traditional TSF and real-world dynamic systems.
- The operationalization of IATSF with the Temporal-Synced IATSF benchmark and the LLM-free FIATS model, whose performance gains are shown to stem from principled influence modeling, not architectural complexity.

## 2 MOTIVATION: TSF FROM SYSTEM ANALYSIS PERSPECTIVE

A fundamental disconnect exists in time series forecasting: while real-world data is generated by dynamic systems shaped by external events, standard models typically operate in a closed loop, using only historical data. This oversight is common even in popular benchmarks like the ETT dataset (Zhou et al., 2021), where crucial external factors like human activity and environmental conditions are ignored even though this system is profoundly affected. While traditional methods like ARIMAX (Majka) can incorporate numerical exogenous variables, they cannot process the rich, qualitative information found in textual sources like news reports or policy updates. Recently, multimodal models have begun to leverage this textual data (Williams et al., 2025; Aksu et al., 2024; Wang et al., 2024a), but their approaches often lack a clear theoretical justification for how influences should be modeled.

To systematically address this qualitative gap, we formally identify and analyze the intrinsic limitations of ignoring qualitative external influences from a dynamical systems perspective[1].

### 2.1 TIME SERIES ARE OBSERVATION OF REAL-WORLD DYNAMIC SYSTEMS

Consider a general dynamical system characterized by hidden states $Z \in \mathbb{R}^m$, evolving based on historical states and independent external influences (Khalil, 2002; Ogata, 2010; Franklin et al., 2010):

$$Z_f = F(Z_h, U_t), \quad X = O(Z) \tag{1}$$

where $F$ represents the true system dynamics, $U_t$ denotes time-varying independent external influences, $O$ represents observation, $X$ for the the observed signal. *For analytical clarity, we assume full observability, i.e. $X = Z$.* We also discuss a simple linear system case $X_f = AX_h + BU_t$,

---

[1] All proofs and discussion are provided in Appendix B, unless otherwise specified.

where $A$ governs self-stimulated state transitions and $B$ encodes influence sensitivity. Standard forecasting datasets $\mathcal{D} = \{(X_h^{(i)}, X_f^{(i)})\}_{i=1}^N$ are generated through sliding window on the observed signals, where $X_h, X_f$ stand for look-back window and forecasting horizon segment accordingly.

## 2.2 THE IMPLICIT SELF-STIMULATION ASSUMPTION IN TSF

Traditional forecasting adopts a *self-stimulation* paradigm where models $f_\theta$ attempt to approximate system dynamics using only historical observations:

$$f_\theta^* = \arg\min_\theta \mathbb{E}\left[\|\epsilon\|^2\right] = \arg\min_\theta \mathbb{E}\left[\|F(X_h, U_t) - f_\theta(X_h)\|^2\right] \tag{2}$$

The critical limitation stems from implicitly treating unobserved influences as hidden random variables $U_t \sim \mathcal{P}_U$. This induces an irreducible forecasting error, as formalized by our first proposition:

**Proposition 2.1** (Self-Stimulation Error Bound). *For any self-stimulated model $f_\theta$, it converges to predicting conditional expectation $F^*(X_h, \mu) \triangleq \mathbb{E}_U[F(X_h, U)]$, the prediction error covariance satisfies:*

$$Cov(\epsilon) \succeq \mathbb{E}_{X_h}\left[\nabla_U F \Sigma (\nabla_U F)^\top\right] \tag{3}$$

*where $\mu = \mathbb{E}(U_t), \quad \Sigma = Cov(U_t)$. For linear systems, this falls back to:*

$$Cov(\epsilon) \succeq B\Sigma B^\top \tag{4}$$

Proposition 2.1 reveals two fundamental limitations: 1) Self-stimulated models converge to predicting *conditional expectations*, rather than true dynamics, explaining prevalent averaging effects in practice as shown in Fig. 1, and 2) An irreducible error floor exists due to influence stochasticity. This establishes a theoretical performance ceiling for conventional TSF approaches.

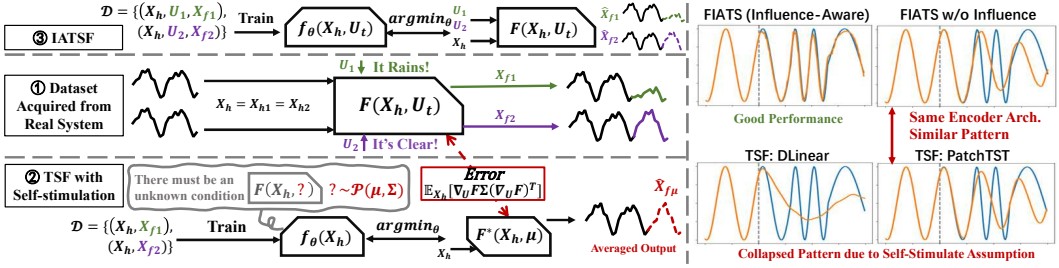

Figure 1: The real system runs under various influences. The influence-Aware method can effectively approximate the real system according to the dataset while traditional self-stimulated method can only approximate a average scenario with persistent error, lead to bad or even collapse result. The right panel shows visualization result of a frequency modulated system which is very sensitive to the influence, i.e. large $\nabla_U F$.

## 3 IATSF: INFLUENCE-AWARE TIME SERIES FORECASTING

### 3.1 TASK FORMULATION

We propose Influence-Aware Time Series Forecasting (IATSF) to overcome the self-stimulation limitation. The key innovation lies in explicit influence modeling:

$$f_\theta^* = \arg\min_\theta \mathbb{E}\left[\|F(X_h, U_t) - f_\theta(X_h, U_t)\|^2\right] \tag{5}$$

where $U_t$ represents measurable influences. This paradigm enables breaking the error bound in proposition 2.1 through influence-aware learning, as detailed in Fig. 1.

As shown above, instead of assuming the external influence stays the same in the TSF, IATSF aims to predict a *conditioned future* with the observed or predicted influence even though it is not fully observed or precise. The error reduction mechanism is formalized through our second proposition:

**Proposition 3.1** (Partial influence Efficacy). *For a system with $p$ independent influences $U_t = \sum_{i=1}^p U_t^i$, incorporating any known influence $U_t^j$ reduces the error covariance by:*

$$\Delta Cov(\epsilon) = \nabla_{U_j} F \Sigma_j (\nabla_{U_j} F)^\top \tag{6}$$

*For linear systems, this reduces the lower bound by $B_j \Sigma_j B_j^\top$.*

Proposition 3.1 demonstrates that *any measurable influence information* reduces forecasting uncertainty, even with incomplete influence knowledge. This motivates our key insight: textual descriptions of influences provide viable information for uncertainty reduction, despite non-numeric formats.

## 3.2 LANGUAGE AS AN INFLUENCE MODALITY

Incorporating exogenous variables is a common approach (Arango et al., 2025; Wang et al., 2024b), but it typically requires numerical time series or one-hot encoded inputs sampled at the same rate as the target series—even when the actual influences are sparse. This limits flexibility, especially when new events occur. In real-world settings, many impactful factors—such as weather anomalies, geopolitical shifts, or human decisions—are hard to quantify but still essential for accurate forecasting. To address this, we propose modeling influences using linguistic descriptors, which naturally capture compositional and relational semantics through lexical encoding. This allows for expressive representations of complex events (e.g., "simultaneous port strikes and agricultural subsidies") without incurring combinatorial overhead. This design offers several key advantages:

**Expert Knowledge Integration**: Textual interfaces facilitate the direct inclusion of domain-specific expertise via natural language specifications (e.g., "anticipated regulatory changes will suppress industrial output"). This makes it easier to incorporate human input or LLM-driven forecasting through linguistic conditioning of influences.

**Generalizability**: Textual representations provide flexibility across various contexts, allowing models to generalize more effectively to new or unseen influence scenarios. The use of natural language reduces reliance on rigid, pre-encoded numerical data, enabling better adaptability to diverse situations.

**Cross-Modal Influence-Modulating**: By embedding both linguistic influence descriptors and their temporal effects in a shared space, neural architectures can learn latent mappings that help modulate the forecasting according to the influence.

# 4 IATSF BENCHMARK

## 4.1 LEAK-FREE DATASET DESIGN

The IATSF benchmark is explicitly constructed to be leak-free, adhering to the principle that models must not access future system states. To enforce this, we only include **independently** evolving influences—external factors that influence the system but are not themselves outcomes of it. Including variables that directly describe or summarize the time series trajectory (as in (Liu et al., 2024a; Jin et al., 2023)) would violate this principle by introducing future state information; see Appendix N for further discussion.

Since system responses to influences often occur much faster than the sampling interval (e.g., photovoltaic panels react to sunlight in milliseconds), *we assume influences take effect instantaneously and denote the up-to-date influence as $U_f$*. In deployment, ground-truth future influences are unavailable, so our benchmark restricts inputs to: (1) **Known information** (e.g., holidays); (2) **Predictions of $U_f$** from expert sources (e.g., weather reports); and (3) **Hypothetical events** for "what-if" scenario analysis. Evaluation strategies accounting for prediction errors in influences are detailed in Appendix B.3.

## 4.2 IATSF DATASETS

Each instance in IATSF is defined as $\mathcal{D} = \left\{ ((X_h^{(i)}, U_f^{(i)}, D), X_f^{(i)}) \right\}_{i=1}^{N}$, comprising historical time series $X_h$, future-aligned influences $U_f$, channel descriptors $D$, and the ground truth future $X_f$. The primary challenge in creating such a benchmark is sourcing influences that are both time-synced and truly independent of the system's state, a requirement that makes standard datasets like ETT (Zhou et al., 2021) unsuitable. Our benchmark addresses this gap by providing datasets across three distinct categories designed for IATSF validation, with full details in Appendix O.

**Toy Systems for Theoretical Validation** This category provides a controlled environment to isolate the impact of influences and empirically verify our theoretical propositions without the noise of complex real-world dynamics. It includes: (1) **Frequency Modulated Toy**, a fully synthetic system where influences precisely control signal frequency, offering a theoretical error bound of zero for

a perfect model; and (2) **Electricity Utility**, which uses real-world appliance data augmented with simple, discrete textual influences like holidays to test the model on basic, real-world patterns.

**Complex Real-World Systems** To test our paradigm on more challenging real-world problems, we use two complex systems where forecasts can be aided by actual influential information. To ensure the external factor is independent and easily obtainable, we use publicly available weather forecasts as the influence. We then evaluate forecasting performance on two distinct systems whose dynamics are affected by weather: (1) **Atmospheric Physics**, which is an ideal system for this task as its variables (e.g., solar radiation, air pressure, dew point) are intrinsically linked to the weather condition. For example, a forecast of "clear skies" allows an IATSF model to infer high solar radiation, a connection a self-stimulated TSF model cannot make. (2) **NYC Traffic Speed**, where the link is less direct but still significant. Urban traffic is potentially influenced by weather; for instance, a "heavy rain" forecast may correlate with slower traffic due to reduced visibility and slick roads. This tests the model's ability to extract more subtle, correlational signals from the influence text.

**Human-Driven Business Systems** This category evaluates IATSF's ability to model volatile market dynamics where historical patterns are often unreliable due to specific events (e.g., a product update). Textual influences are often the primary signal for future performance, especially for new products with limited history (the 'cold-start' problem). The **Game Active User Dataset (GAUD)** tracks daily active users for 90 games, with developer logs as influences, testing the model's practical utility for business decision-making under uncertainty.

## 5 FIATS: A SIMPLE SYSTEM-AWARE BASELINE MODEL FOR IATSF

Having established through control theory that breaking the 'self-stimulation' barrier is essential for forecasting progress, we now introduce the model designed to achieve this. We propose FIATS, the architectural embodiment of our **Influence-Aware Time Series Forecasting (IATSF)** paradigm. While recent studies (Aksu et al., 2024; Williams et al., 2025; Liu et al., 2024a; Wang et al., 2024a; Niu et al., 2025) explore text-informed forecasting with large language models (LLMs), their architectural complexity and significant overhead obscure whether performance gains stem from genuine influence modeling or simply from increased model capacity. To provide a rigorous and interpretable validation of the IATSF framework, we designed FIATS as the first LLM-free, numerical-based forecaster built from first principles for this task. As illustrated in Fig. 2, FIATS integrates a standard patch-based time series encoder with a novel influence semantic encoder and decoder that operate on text embeddings directly, avoiding the variance and token overhead associated with generative LLMs. The novelty is as follows:

**Temporal-Synced Influence** Real-world systems often respond rapidly to influences, necessitating temporal alignment between text and time series data. FIATS addresses this by synchronizing each time series patch with the last influence observed, e.g. for patch start from 10:15, sync with the last timestep with influence update of 10:00. This ensures the model uses only *leak-free, contemporaneous* influences when forecasting subsequent patches, preventing future information leakage while maintaining temporal relevance.

**Channel-aware Adaptive Sensitivity Modeling (CASM)** Proposition 3.1 shows that the error reduction depends on the system's sensitivity to the influence. CASM is designed to explicitly model this sensitivity for each channel, learning how a given influence (e.g., 'clear skies') should affect different time series (e.g., solar radiation vs. atmospheric pressure). Starting from linear systems where time series are observed by $X_f = CZ_f = CAZ_h + CBU_f$, channel-specific sensitivity to influences is governed by $\frac{dx_f^i}{dU_f} = c^i B$. This indicates that each channel responds differently to external influences. The error analysis is discussed in Appendix B.4. Cross-attention provides an ideal framework for this, as its core mechanism naturally computes a weighted alignment between two sets of inputs—in our case, mapping textual influences (keys) to specific time series channels (queries). Specifically, to capture this without introducing excessive parameters, we introduce Channel-aware Adaptive Sensitivity Modeling Block, as shown in the right panel of the Fig. 2:

- *Query as Channel-wise Sensitivity* $\tilde{C} = Desc \cdot W_Q$: Channel descriptions $Desc \in \mathbb{R}^{CN \times D}$ are served as query ($CN$ as channel number). The query projection explicitly learns how textual channel features (e.g., "atmospheric pressure") influence influence sensitivity for each channel. This allows the model to adjust how influences are perceived based on channel-specific characteristics.

- *Key as influence Filter $\tilde{B}_{U_f} = (News \cdot W_K)^\top$*: The key projection maps temporal-synced news embeddings $News \in \mathbb{R}^{M \times D}$ to a system sensitivity matrix ($M$ as news number), allowing the model to filter out irrelevant influences (e.g., excluding "tech stock news" when forecasting atmospheric physics). This ensures that only pertinent influences are considered for each system.

- *Value as influence Translator $\tilde{U}_f = News \cdot W_V$*: Value projection learns to maps news text embedding to $\tilde{U}_f$, the latent space of actionable influence effects.

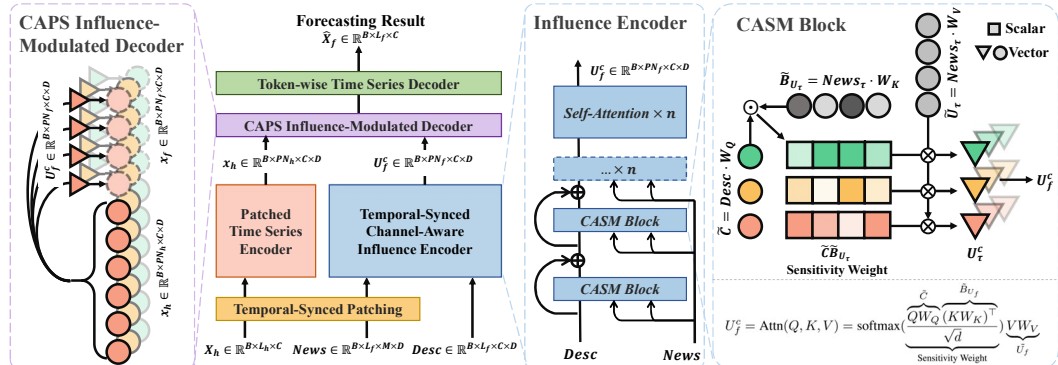

Figure 2: **Architecture of FIATS.** FIATS integrates three inputs: time series data from a look-back window, temporal-synced news embeddings, and channel description embeddings. The influence encoder employs CASM blocks in a residual connection along with multiple self-attention layers to enhance feature extraction. The CAPS influence-modulated decoder projects the historical time series embeddings into the future, guided by channel-aware, time-synced influences. A token-wise decoder is used to prevent overfitting in the final linear layer, as discussed in (Lee et al., 2023).

The above analysis show that the attention mechanism can effectively generate the channel-aware influence $U_f^c$. This design allows identical influences to differentially impact channels based on their descriptions. Unlike static sensitivity coefficients found in classical systems, this formulation maintains the nonlinear characteristics provided by the transformer block, allowing for greater *learning flexibility* to approximate complex nonlinear system. Additionally, it aligns well with the theoretical framework, making the model more *interpretable*. The attention map produced by the CASM layer directly reveals the sensitivity of each channel to various influences, providing clear insights into how influences impact different channels based on their specific descriptions.

**Channel-Aware Parameter Sharing (CAPS)** While CASM addresses heterogeneous influence responses, channels also exhibit inherent differences in their temporal patterns – a critical factor neglected by conventional parameter sharing. Previous shared models approximate all channels with a same set of parameters introducing persistent errors $\epsilon_i = o_i(Z) - \frac{1}{k}\sum_{j=1}^{k} o_j(Z)$ where $o_i$ for real system channel-specific dynamics.

To mitigate this issue, FIATS introduces a lightweight channel-aware decoding mechanism. All channels are first encoded into a shared latent space $\tilde{Z}$ by a unified time-series encoder. Then, a channel-conditioned decoder is used to adaptively project this latent representation into a channel-aware space, conditioned by the channel-specific time-synced influence embeddings $U_f^c$. decoder approximates channel-specific adjustments by modulating the shared latent space through cross-attention $Attention(Q = U_t^c, K, V = \tilde{Z})$ to simulate such nonlinear projection. To avoid future information leakage, we apply causal attention mask here. We will omit the analysis.

This design introduces minimal overhead while enabling the model to account for channel heterogeneity in a flexible, data-driven manner. Additionally, the attention maps produced by the channel-aware decoder are interpretable: they reveal how each channel selectively attends to historical time series data under different influences. We provide visualizations and further analysis of these attention patterns in the following session.

# 6 EXPERIMENTS

**Baseline Models** FIATS is benchmarked against several state-of-the-art (SOTA) methods. These include linear-based models (Zeng et al., 2023; Xu et al., 2023) , transformer-based models (Nie

et al., 2023; Liu et al., 2023), and fine-tuned LLM-based multimodal method (Jin et al., 2023). Additionally, we compare pretrained time series "foundation models" (Ansari et al., 2024; Woo et al., 2024; Shi et al., 2025). This selection covers a range of approaches, including self-stimulated linear and nonlinear models, data-specific and pretrained models, LLM-based cross-modal models.

## 6.1 RQ1: CAN IATSF OVERCOME THE LIMITATIONS OF SELF-STIMULATION?

To directly test our theoretical claims from Section 2, we evaluate FIATS on our two toy systems, which provide controlled environments to isolate the impact of influences. The **FM Toy** dataset offers a theoretical error bound of zero, while the **Electricity Utility** dataset tests the paradigm on real data with simple, discrete influences (e.g., holidays).

**Statistical Results** The results provide strong empirical evidence for our theory. On the FM Toy dataset, Table 1 shows that FIATS achieves a near-zero error, directly approaching the theoretical lower bound. In stark contrast, all self-stimulated TSF methods, including massive pre-trained models, fail spectacularly. This confirms that the performance bottleneck is indeed the flawed "self-stimulation" assumption, not model scale. The visualization in Fig. 1 shows these models producing collapsed, averaged-out forecasts, perfectly aligning with Proposition 2.1. On the Electricity dataset, FIATS again demonstrates SOTA performance by effectively leveraging minimal textual cues. Interestingly, even a powerful model like TimeLLM shows some capability here, suggesting that when influences are simple and causal links are obvious, large models can partially succeed, but our lightweight, principled approach is more effective and efficient.

Table 1: Forecasting result in MSE, comparing the *influence-aware FIATS* against various TSF methods. The best result is highlighted in bold and the second best is highlighted in underscore.

| Dataset | Pred. Len. | FIATS | FITS | DLinear | PatchTST | iTrans. | Chronos-L | MOIRAI-L | Time-MoE-U | TimeLLM |
|---|---|---|---|---|---|---|---|---|---|---|
| FM Toy | 14 | **0.003** | 0.282 | 0.151 | 0.006 | 0.136 | 0.012 | 0.013 | 0.012 | 0.231 |
| | 28 | **0.008** | 0.692 | 0.297 | 0.029 | 0.295 | 0.047 | 0.062 | 0.035 | 0.382 |
| | 60 | **0.020** | 0.909 | 0.442 | 0.075 | 0.494 | 0.129 | 0.133 | 0.107 | 0.551 |
| | 120 | **0.027** | 0.883 | 0.632 | 0.168 | 0.747 | 0.374 | 0.385 | 0.295 | 0.788 |
| Electricity Utility | 96 | **0.124** | 0.134 | 0.140 | 0.130 | 0.148 | 0.154 | 0.152 | 0.149 | 0.131 |
| | 192 | **0.144** | 0.149 | 0.153 | 0.149 | 0.162 | 0.177 | 0.171 | 0.168 | 0.152 |
| | 336 | **0.158** | 0.165 | 0.169 | 0.166 | 0.178 | 0.197 | 0.192 | 0.183 | 0.160 |
| | 720 | **0.190** | 0.203 | 0.204 | 0.210 | 0.225 | 0.242 | 0.236 | 0.229 | 0.192 |
| NYC Traffic Speed | 96 | **0.443** | 0.973 | 0.957 | 0.858 | 0.858 | 0.913 | 0.997 | 0.980 | 0.974 |
| | 192 | **0.609** | 1.161 | 1.123 | 1.031 | 1.026 | 1.217 | 1.272 | 1.250 | 1.232 |
| | 336 | **0.685** | 1.306 | 1.262 | 1.176 | 1.195 | 1.512 | 1.594 | 1.421 | 1.575 |
| | 720 | **0.710** | 1.457 | 1.378 | 1.275 | 1.295 | 1.799 | 1.825 | 1.592 | 1.729 |
| Atmospheric Physics 2014-19 | 96 | **0.182** | 0.248 | 0.294 | 0.252 | 0.267 | 0.293 | 0.299 | 0.258 | 0.294 |
| | 192 | **0.205** | 0.297 | 0.340 | 0.304 | 0.327 | 0.357 | 0.356 | 0.318 | 0.342 |
| | 336 | **0.235** | 0.354 | 0.393 | 0.364 | 0.404 | 0.448 | 0.457 | 0.413 | 0.393 |
| | 720 | **0.281** | 0.430 | 0.456 | 0.439 | 0.495 | 0.512 | 0.532 | 0.508 | 0.461 |
| Atmospheric Physics 2014-24 | 96 | **0.410** | 0.436 | 0.487 | 0.464 | 0.456 | 0.447 | 0.453 | 0.437 | - |
| | 192 | **0.438** | 0.524 | 0.568 | 0.567 | 0.578 | 0.552 | 0.557 | 0.542 | - |
| | 336 | **0.455** | 0.601 | 0.644 | 0.644 | 0.698 | 0.685 | 0.673 | 0.647 | - |
| | 720 | **0.497** | 0.692 | 0.725 | 0.745 | 0.832 | 0.754 | 0.765 | 0.734 | - |

## 6.2 RQ2: DOES IATSF EXCEL IN COMPLEX REAL-WORLD SYSTEMS?

Having validated our theory in controlled settings, we now investigate if the IATSF paradigm provides meaningful gains in noisy, complex systems. We use two distinct datasets, **Atmospheric Physics** and **NYC Traffic Speed**, which are both potentially correlated with the weather condition — an independent influence. The Atmospheric Physics system has strong, direct physical links to weather, while the NYC Traffic system has a more subtle, indirect relationship. This setup allows us to test whether FIATS can successfully extract these different types of correlations from the training data and leverage them to outperform self-stimulated models that are blind to this external context.

**Statistical Results** The results confirm FIATS's ability to capitalize on external information. As shown in Table 1, FIATS consistently outperforms all baselines, achieving an average MSE reduction of 36.0% on Atmospheric Physics and 44.3% on NYC Traffic Speed compared to the strongest self-stimulated baseline, PatchTST. This performance gap highlights that even for complex systems, providing external context is critical. Pretrained models like Chronos-L, despite their vast training data, underperform FIATS, underscoring that scaling data alone cannot compensate for missing, crucial influence information. Table 2 further breaks down performance by channel, showing that FIATS achieves substantial gains even on variables not directly mentioned in the weather reports (e.g., pressure p, air density $\rho$, and vapor pressure VPdef), demonstrating its ability to infer latent correlation.

**Case Study: Visualization and Controllability** To understand why FIATS excels, Fig. 3 visualizes three representative channels from the Atmospheric Physics dataset. The first channel, atmospheric pressure (p), is sensitive to regional climate shifts but lacks

Table 2: A selection of channel-wise performance on Atmos. Phy. 2014-19 dataset in MSE.

| Channel | FIATS | FITS | DLinear | PatchTST | iTrans. | IMP. |
|---|---|---|---|---|---|---|
| p (mbar) | **0.136** | 0.863 | 0.823 | 0.930 | 1.032 | **83.43%** |
| Tpot (K) | **0.182** | 0.316 | 0.352 | 0.322 | 0.353 | 42.18% |
| VPdef (mbar) | **0.283** | 0.638 | 0.696 | 0.674 | 0.803 | **55.59%** |
| rho (g/m³) | **0.192** | 0.390 | 0.411 | 0.418 | 0.453 | **50.73%** |
| raining (s) | **0.790** | 0.873 | 0.937 | 0.859 | 0.994 | 8.04% |
| SWDR (W/m²) | **0.182** | 0.308 | 0.385 | 0.296 | 0.377 | 38.39% |

strong short-term historical correlation. Its slow, subtle changes are challenging for traditional TSF models. PatchTST fails to capture these dynamics, defaulting to a flat prediction, while FIATS successfully models the trend by conditioning on relevant influences. The second channel, rainfall duration, is sparse and lacks periodicity. PatchTST outputs near-zero values—its conditional expectation under uncertainty—while FIATS adjusts its predictions based on available influence signals. It correctly forecasts the first rainfall event but misses the second due to misaligned or absent external information, reflecting a candid dependence on accurate influence input. The third channel, solar radiation (SWDR), is not explicitly mentioned in the influence but is indirectly implied. FIATS captures its phase and amplitude accurately, thanks to the CASM design that enables cross-channel sensitivity modeling. PatchTST, by comparison, produces generic, misaligned waveforms.

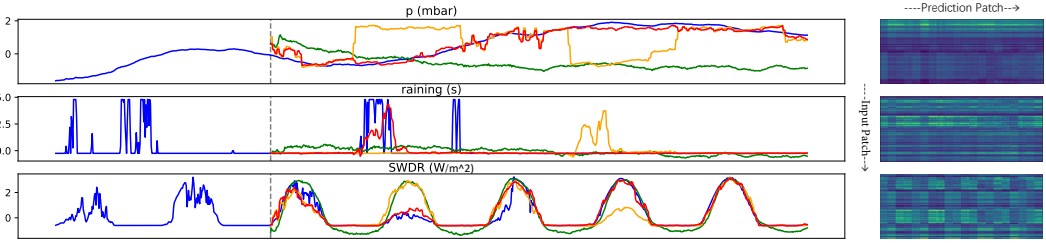

Figure 3: Visualization of three channels on the 15,000th test sample of the Atmos. Phy. 2014-19 dataset. Blue indicates ground truth, Red shows FIATS, Green represents PatchTST, and Orange denotes FIATS with swapped influences on the second and fourth forecast days. The CAPS influence-modulated decoder exhibits distinct attention patterns across channels.

### 6.3 RQ3: How Does IATSF Handle Human-Driven Market Dynamics?

We next evaluate IATSF's ability to model systems driven by human decisions and external events using the **GAUD** dataset. This scenario presents unique challenges, including high variability and cold-start problems for new games, where historical data is sparse but influence information (e.g., developer logs, marketing) is available.

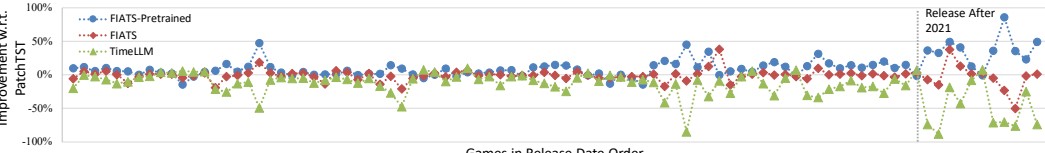

Figure 4: Performance improvement with respect to the PatchTST on each time series in GAUD.

As shown in Fig. 4, FIATS consistently outperforms PatchTST, achieving an average improvement of 12.6% and ranking first on 59.6% of the games. The advantage is most pronounced for games released after 2021, where short time series cause traditional models to fail. FIATS's ability to generalize from textual influences allows it to deliver robust forecasts even in these data-scarce, cold-start scenarios. This demonstrates the paradigm's practical utility for business decision-making and market analysis.

### 6.4 RQ4: What Makes the FIATS Architecture Effective and Robust?

Finally, we conduct a series of ablations and analyses to verify that the observed performance gains stem from our principled architectural design rather than confounding factors.

**Interpretability via Attention Maps** The CASM block analysis in Fig. 5 shows how the model focuses on different temporal features across layers. In the first layer, attention centers on the first sentence, providing temporal context for daily and annual periodicity. The second layer shifts attention to channel-specific signals, particularly the sixth sentence describing atmospheric pressure, reflecting the model's sensitivity to channel-specific patterns and influences. By the third layer, attention

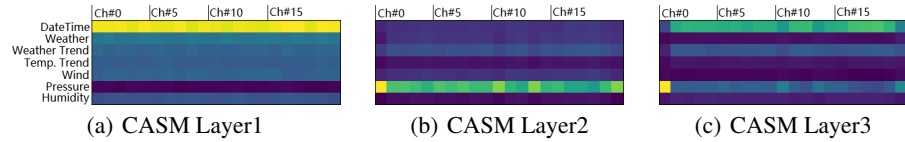

(a) CASM Layer1        (b) CASM Layer2        (c) CASM Layer3

Figure 5: Attention map of the CASM (i.e. sensitivity weight) on the 15000th test sample of Atmos. Phy. 2014-19 dataset. We use three cross attention block in residual connection. The horizontal axis stands for channels and vertical stands for the 7 sentences of the weather report summary.

diversifies, focusing on relevant influence aspects for each channel. The CAPS influence-modulated decoder, shown in Fig. 3, demonstrates distinct attention patterns across channels, highlighting the model's ability to align time series data with textual influences. Channels associated with periodic variables like SWDR exhibit clear periodicity in attention maps, indicating effective capture of cyclical patterns. The rainfall channel highlights historical rainfall, showcasing the model's sensitivity to key moments. This adaptability is driven by CASM, enabling the model to tailor its attention based on each channel's unique characteristics.

Table 3: Ablation result on Atmos. Phy. 2014-19 in MSE.

| Pred. Len. | Openai 512 | MiniLM | mpnet | Zero Desc. | Zero News |
|---|---|---|---|---|---|
| 96 | **0.182** | 0.186 | 0.196 | 0.209 | 0.249 |
| 192 | **0.205** | 0.214 | 0.216 | 0.260 | 0.302 |
| 336 | 0.235 | **0.232** | 0.251 | 0.302 | 0.359 |
| 720 | 0.281 | **0.272** | 0.291 | 0.356 | 0.432 |

**Ablation Studies** We first prove the necessity of our core components. As shown in Table 3, when we remove influence inputs entirely ("Zero News"), performance drops to that of a self-stimulated model, proving that the gains come from the influences themselves. Crucially, removing channel descriptions ("Zero Desc.") also significantly degrades performance, confirming the critical role of the CASM mechanism in modeling channel-specific sensitivities. Next, we test how the quality of influences affects the model. Fig. 6 shows that while FIATS is robust to minor semantic noise, performance degrades as influence inputs become less accurate, a finding that directly supports Proposition 3.1. Finally, we confirm the architecture's generalizability by swapping text embedding models; Table 3 shows that performance remains stable across different embedding spaces.

**Summary of Findings** Our experiments decisively validate the IATSF paradigm. Controlled experiments (RQ1) confirm our control-theoretic analysis, showing FIATS approaches the theoretical error bound while even the largest foundation models fail without influence data. This success extends to complex real-world systems (RQ2, RQ3), where FIATS consistently outperforms SOTA baselines. Crucially, architectural analyses (RQ4) attribute these gains to our principled design choices—CASM and CAPS—not model scale. The results demonstrate that IATSF, operationalized through the interpretable and robust FIATS model, represents a validated, theoretically-grounded, and efficient path forward for the field.

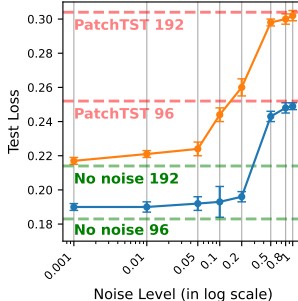

Figure 6: Loss under various noise levels. Blue line for horizon 96 and Orange line for horizon 192.

# 7 CONCLUSION, LIMITATION & FUTURE WORK

This paper presents Influence-Aware Time Series Forecasting (IATSF), leveraging a control-theoretic framework to address errors from the self-stimulation assumption and improve forecasting accuracy through influence modeling. We demonstrate the effectiveness of IATSF using the Temporal-Synced IATSF benchmark and the FIATS model, which outperforms state-of-the-art methods, including those based on large language models. Our findings emphasize that influence-aware modeling, rather than simply increasing model complexity, is crucial for enhancing forecasting performance. While FIATS shows some capability in noise tolerance and generalization, challenges persist in modeling complex chaotic systems, where influences may not have immediate effects and varying credibility of news sources or temporal misalignment could lead to inaccurate influence observations. Overcoming these challenges will require more advanced models, potentially benefiting from pretraining techniques. These areas will be explored in future research. Additionally, the analysis framework can inspire further exploration, such as modeling multichannel correlation.

BROADER IMPACT, ETHIC STATEMENT AND CODE AVAILABILITY

This paper presents work whose goal is to advance the field of Machine Learning. There are many potential societal consequences of our work, none which we feel must be specifically highlighted here.

We comply with intellectual property agreements for all data sources. Data are properly anonymized and content generated by OpenAI API is free for general use, with no concerns regarding sensitive or illegal activity in our dataset.

The code for TGForecaster and dataset samples are available at: `https://anonymous.4open.science/r/IATSF_review-F624`.

LLM USAGE STATEMENT

We use Large Language Models (LLMs), including ChatGPT and Gemini, solely for polishing the writing of this paper.

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

# A    SOME NOTATION USED IN PAPER

Table 4: Summary of Some Notations Used in the Paper

| Symbol | Description |
|---|---|
| $F$ | Real System dynamics function |
| $\theta$ | Parameters of the forecasting model |
| $f_\theta$ | Forecasting function with parameters $\theta$ |
| $O$ | Observation function |
| $Z$ | Hidden states of the system |
| $X$ | Observed signal from the system |
| $X_f$ | Future time Series Segment |
| $X_h$ | Historical time Series Segment |
| $\hat{X}_f$ | Forecasted future time series |
| $U_t$ | Time varying external influence |
| $U_f$ | External influence for the future segment |
| $\Sigma$ | Covariance of the influences $U_t$ |
| $\mu$ | Mean of the influence distribution |
| $\mathcal{D}$ | Set of time series samples |
| $W_Q, W_K, W_V$ | Weights for Query, Key, and Value in the attention mechanism |

# B    PROOF AND DISCUSSION

In this section, we give proof of the two proposition mentioned in the paper. We also discuss the error introduced by the external influence forecaster, weight sharing and incomplete observation.

## B.1    PROPOSITION 2.1: ERROR BOUND INTRODUCED BY SELF-STIMULATION

### B.1.1    MOST SIMPLE CASE: LINEAR SYSTEM, LINEAR MODEL

*Proof.* Consider a linear system with unobserved influences $U$:

$$X_f = AX_h + BU, \quad U \sim \mathcal{P}_U \text{ (i.i.d.)}, \ \mathbb{E}[U] = \mu, \ \text{Cov}(U) = \Sigma, \tag{7}$$

where $X_h$ represents historical states and $X_f$ represents future states. We aim to estimate $X_f$ using a self-stimulated linear model:

$$\hat{X}_f = CX_h + d, \tag{8}$$

where $C$ and $d$ are parameters to be estimated via least squares by minimizing the loss:

$$\mathcal{L}(C, d) = \mathbb{E}\left[\|X_f - (CX_h + d)\|^2\right]. \tag{9}$$

To find the optimal parameters $C^*$ and $d^*$, we take derivatives with respect to $d$ and $C$ and set them to zero.

Taking the derivative with respect to $d$:

$$\frac{\partial \mathcal{L}}{\partial d} = \mathbb{E}\left[-2(X_f - CX_h - d)\right] = 0$$

$$\mathbb{E}[d] = \mathbb{E}[X_f - CX_h]$$

So, for the optimal $C^*$, $d^*$ is:

$$d^* = \mathbb{E}[X_f] - C^*\mathbb{E}[X_h]. \tag{10}$$

Substituting $X_f = AX_h + BU$:

$$d^* = \mathbb{E}[AX_h + BU] - C^*\mathbb{E}[X_h] = A\mathbb{E}[X_h] + B\mathbb{E}[U] - C^*\mathbb{E}[X_h]$$

$$d^* = (A - C^*)\mathbb{E}[X_h] + B\mu.$$

Next, taking the (Fréchet) derivative with respect to $C$ (or considering element-wise derivatives $\frac{\partial \mathcal{L}}{\partial C_{ij}}$), we set $\nabla_C \mathcal{L}(C, d^*) = 0$:

$$\mathbb{E}\left[\nabla_C \|X_f - CX_h - d^*\|^2\right] = 0$$

$$\mathbb{E}\left[-2(X_f - CX_h - d^*)X_h^\top\right] = 0$$

$$\mathbb{E}\left[(X_f - CX_h - d^*)X_h^\top\right] = 0$$

Substituting $X_f = AX_h + BU$ and $d^* = (A - C)\mathbb{E}[X_h] + B\mu$:

$$\mathbb{E}\left[(AX_h + BU - CX_h - ((A - C)\mathbb{E}[X_h] + B\mu))X_h^\top\right] = 0$$

$$\mathbb{E}\left[((A - C)X_h + B(U - \mu) - (A - C)\mathbb{E}[X_h])X_h^\top\right] = 0$$

$$\mathbb{E}\left[(A - C)(X_h - \mathbb{E}[X_h])X_h^\top\right] + \mathbb{E}\left[B(U - \mu)X_h^\top\right] = 0$$

Assuming $X_h$ and $U$ are independent (or at least $U - \mu$ is uncorrelated with $X_h$), $\mathbb{E}[(U - \mu)X_h^\top] = \mathbb{E}[U - \mu]\mathbb{E}[X_h^\top] = 0 \cdot \mathbb{E}[X_h^\top] = 0$.

$$(A - C)\mathbb{E}\left[(X_h - \mathbb{E}[X_h])X_h^\top\right] = 0$$

$$(A - C)\left(\mathbb{E}[X_h X_h^\top] - \mathbb{E}[X_h]\mathbb{E}[X_h^\top]\right) = 0$$

$$(A - C)\text{Cov}(X_h) = 0.$$

If $\text{Cov}(X_h)$ is invertible (i.e., its columns are linearly independent and it has full rank), then we must have $A - C = 0$, which implies:

$$C^* = A. \tag{11}$$

Substituting $C^* = A$ back into the expression for $d^*$:

$$d^* = (A - A)\mathbb{E}[X_h] + B\mu = B\mu. \tag{12}$$

So the optimal parameters are:

$$C^* = A, \quad d^* = B\mu. \tag{13}$$

The prediction error is given by:

$$\epsilon = X_f - \hat{X}_f = (AX_h + BU) - (C^*X_h + d^*) = (A - C^*)X_h + BU - d^*. \tag{14}$$

Substituting the optimal parameters $C^* = A$ and $d^* = B\mu$:

$$\epsilon = (A - A)X_h + BU - B\mu = B(U - \mu). \tag{15}$$

The mean of the error is $\mathbb{E}[\epsilon] = \mathbb{E}[B(U - \mu)] = B(\mathbb{E}[U] - \mu) = B(\mu - \mu) = 0$. The covariance of the error is:

$$\text{Cov}(\epsilon) = \mathbb{E}\left[\epsilon\epsilon^\top\right] = \mathbb{E}\left[(B(U - \mu))(B(U - \mu))^\top\right] \tag{16}$$

$$\text{Cov}(\epsilon) = \mathbb{E}\left[B(U - \mu)(U - \mu)^\top B^\top\right] = B\mathbb{E}\left[(U - \mu)(U - \mu)^\top\right]B^\top$$

Since $\text{Cov}(U) = \Sigma = \mathbb{E}\left[(U - \mu)(U - \mu)^\top\right]$,

$$\text{Cov}(\epsilon) = B\Sigma B^\top. \tag{17}$$

This covariance represents the irreducible error floor caused by the unobserved influence $U$.

Let $F(X_h, U) = AX_h + BU$ be the true underlying system for $X_f$. The gradient of $F$ with respect to $U$ is $\nabla_U F = B$. Thus, even with optimal parameters, the error covariance satisfies:

$$\mathbb{E}[\epsilon\epsilon^\top] = B\Sigma B^\top = (\nabla_U F)\Sigma(\nabla_U F)^\top. \tag{18}$$

If we consider a general form of a lower bound related to the influence of $U$, such as one involving an expectation over $X_h$, $\mathbb{E}_{X_h}\left[(\nabla_U F)\Sigma(\nabla_U F)^\top\right]$, in this linear case it simplifies directly to $B\Sigma B^\top$ because $\nabla_U F = B$ does not depend on $X_h$. Therefore:

$$\mathbb{E}[\epsilon\epsilon^\top] \succeq \mathbb{E}_{X_h}\left[(\nabla_U F)\Sigma(\nabla_U F)^\top\right] \tag{19}$$

where $\succeq$ denotes positive semi-definiteness (Löwner order). In this specific linear case, this holds with equality: $\mathbb{E}[\epsilon\epsilon^\top] = B\Sigma B^\top$. This lower bound arises from the unobserved influence $U$. $\qquad\square$

### B.1.2 A Step Further: Linear System, Nonlinear Model

*Proof.* Consider the same linear system with unobserved influences $U$:

$$X_f = AX_h + BU, \quad U \sim \mathcal{P}_U \text{ (i.i.d.)}, \ \mathbb{E}[U] = \mu, \ \text{Cov}(U) = \Sigma. \tag{20}$$

We now use a nonlinear self-stimulated model (e.g., an arbitrary machine learning model) for prediction:

$$\hat{X}_f = f(X_h). \tag{21}$$

The optimal self-stimulated model $f_{\text{opt}}(X_h)$ that minimizes the Mean Squared Error (MSE) is the conditional expectation of $X_f$ given $X_h$:

$$f_{\text{opt}}(X_h) = \mathbb{E}[X_f \mid X_h] = \mathbb{E}[AX_h + BU \mid X_h].$$

Assuming $U$ is independent of $X_h$ ($U \perp X_h$), then $\mathbb{E}[U \mid X_h] = \mathbb{E}[U] = \mu$. So,

$$f_{\text{opt}}(X_h) = AX_h + B\mu.$$

The prediction error is $\epsilon = X_f - f(X_h)$. Substituting $X_f$:

$$\epsilon = (AX_h + BU) - f(X_h).$$

We can rewrite this by adding and subtracting $B\mu$:

$$\epsilon = \underbrace{(AX_h + B\mu - f(X_h))}_{\text{Model Inadequacy Term } \Delta_f(X_h)} + \underbrace{B(U - \mu)}_{\text{Zero-Mean Stochastic Term}}. \tag{22}$$

Let $\Delta_f(X_h) = (AX_h + B\mu - f(X_h))$. This term represents how well the model $f(X_h)$ approximates the optimal predictor $f_{\text{opt}}(X_h)$. The stochastic term $B(U - \mu)$ has $\mathbb{E}[B(U - \mu)] = 0$.

The mean of the error is $\mathbb{E}[\epsilon] = \mathbb{E}[\Delta_f(X_h)]$. For an unbiased $f(X_h)$ relative to $f_{opt}(X_h)$, $\mathbb{E}[\Delta_f(X_h)] = 0$. The covariance of the error is $\text{Cov}(\epsilon) = \mathbb{E}[(\epsilon - \mathbb{E}[\epsilon])(\epsilon - \mathbb{E}[\epsilon])^\top]$. If we assume $\mathbb{E}[\Delta_f(X_h)] = 0$ (i.e., $f(X_h)$ is unbiased for $AX_h + B\mu$ on average), then $\mathbb{E}[\epsilon] = 0$. The error covariance is:

$$\text{Cov}(\epsilon) = \mathbb{E}[\epsilon\epsilon^\top] = \mathbb{E}\left[(\Delta_f(X_h) + B(U - \mu))(\Delta_f(X_h) + B(U - \mu))^\top\right].$$

Expanding this:

$$\text{Cov}(\epsilon) = \mathbb{E}[\Delta_f(X_h)\Delta_f(X_h)^\top] + \mathbb{E}[\Delta_f(X_h)(U-\mu)^\top B^\top] + \mathbb{E}[B(U-\mu)\Delta_f(X_h)^\top] + \mathbb{E}[B(U-\mu)(U-\mu)^\top B^\top].$$

Since $U \perp X_h$, $\Delta_f(X_h)$ (a function of $X_h$) is independent of $U - \mu$. Thus, the cross-terms are zero:

$$\mathbb{E}[\Delta_f(X_h)(U-\mu)^\top B^\top] = \mathbb{E}[\Delta_f(X_h)]\mathbb{E}[(U-\mu)^\top]B^\top = \mathbb{E}[\Delta_f(X_h)] \cdot 0 \cdot B^\top = 0.$$

So, the error covariance becomes:

$$\text{Cov}(\epsilon) = \underbrace{\mathbb{E}[\Delta_f(X_h)\Delta_f(X_h)^\top]}_{\text{MSE of model inadequacy}} + B\Sigma B^\top. \tag{23}$$

The term $\mathbb{E}[\Delta_f(X_h)\Delta_f(X_h)^\top]$ is the mean squared error of $f(X_h)$ in approximating $AX_h + B\mu$. This term is always positive semi-definite. Therefore,

$$\text{Cov}(\epsilon) \succeq B\Sigma B^\top.$$

This means $B\Sigma B^\top$ is an irreducible lower bound on the error covariance, regardless of the complexity of $f(X_h)$, as long as $f(X_h)$ only uses $X_h$. If $f(X_h)$ perfectly fits the optimal deterministic component, i.e., $f(X_h) = f_{\text{opt}}(X_h) = AX_h + B\mu$, then $\Delta_f(X_h) = 0$. The error reduces to:

$$\epsilon = B(U - \mu). \tag{24}$$

The covariance of this minimal error is:

$$\text{Cov}(\epsilon) = B\Sigma B^\top. \tag{25}$$

Any claim that a model $f'(X_h)$ achieves $\text{Cov}(\epsilon) \prec B\Sigma B^\top$ would imply that $\mathbb{E}[\Delta_{f'}(X_h)\Delta_{f'}(X_h)^\top]$ in Eq. equation 23 would have to be negative definite, which is impossible as it is a matrix of expected outer products (a sum of positive semi-definite matrices) which contradict with the independent influence assumption. $\square$

### B.1.3 REAL SCENARIO: NONLINEAR MODEL, NONLINEAR SYSTEM

*Proof.* Consider a general nonlinear system:

$$X_f = F(X_h, U), \quad U \sim \mathcal{P}_U \text{ (i.i.d.)}, \ \mathbb{E}[U] = \mu, \ \text{Cov}(U) = \Sigma, \tag{26}$$

where $F$ is a nonlinear state transition function. We use a self-stimulated model:

$$\hat{X}_f = f(X_h), \tag{27}$$

where $f$ is an arbitrary nonlinear model.

The optimal model $f_{\text{opt}}(X_h)$ that minimizes MSE is $\mathbb{E}[X_f \mid X_h]$. Assuming $U \perp X_h$:

$$f_{\text{opt}}(X_h) = \mathbb{E}_U[F(X_h, U) \mid X_h] = \mathbb{E}_U[F(X_h, U)].$$

Let $F^*(X_h) = \mathbb{E}_U[F(X_h, U)]$. The prediction error is:

$$\epsilon = X_f - f(X_h) = \underbrace{(F^*(X_h) - f(X_h))}_{\text{Model Inadequacy}} + \underbrace{(F(X_h, U) - F^*(X_h))}_{\text{Irreducible Stochastic Error}}. \tag{28}$$

The term $F(X_h, U) - F^*(X_h)$ has zero mean conditional on $X_h$ (and thus zero unconditional mean). The Model Inadequacy term $F^*(X_h) - f(X_h)$ reflects how well $f(X_h)$ approximates the true conditional mean $F^*(X_h)$.

The covariance of the error, assuming $\mathbb{E}[F^*(X_h) - f(X_h)] = 0$, is:

$$\text{Cov}(\epsilon) = \mathbb{E}[(F^*(X_h) - f(X_h))(F^*(X_h) - f(X_h))^\top] + \mathbb{E}[(F(X_h, U) - F^*(X_h))(F(X_h, U) - F^*(X_h))^\top].$$

The cross-terms vanish due to the independence of $U$ from $X_h$ and the property that $\mathbb{E}_U[F(X_h, U) - F^*(X_h) \mid X_h] = 0$. The first term is positive semi-definite. So,

$$\text{Cov}(\epsilon) \succeq \mathbb{E}[(F(X_h, U) - F^*(X_h))(F(X_h, U) - F^*(X_h))^\top] = \mathbb{E}_{X_h}[\text{Cov}_U(F(X_h, U) \mid X_h)].$$

The term $\text{Cov}_U(F(X_h, U) \mid X_h)$ is the conditional variance of $F(X_h, U)$ given $X_h$. Using a first-order Taylor expansion for $F(X_h, U)$ around $U = \mu$: $F(X_h, U) \approx F(X_h, \mu) + \nabla_U F(X_h, \mu)(U - \mu)$. Then $F^*(X_h) = \mathbb{E}_U[F(X_h, U)] \approx F(X_h, \mu) + \nabla_U F(X_h, \mu)\mathbb{E}_U[U - \mu] = F(X_h, \mu)$, neglecting higher-order terms (e.g., terms like $\frac{1}{2}\text{Tr}(\Sigma H_F)$ where $H_F$ is the Hessian w.r.t $U$). Under this approximation, the irreducible stochastic error is $F(X_h, U) - F^*(X_h) \approx \nabla_U F(X_h, \mu)(U - \mu)$. The conditional error covariance, given $X_h$, is approximately:

$$\text{Cov}(\epsilon \mid X_h; f = f_{\text{opt}}) \approx \mathbb{E}_U[\nabla_U F(X_h, \mu)(U - \mu)(U - \mu)^\top \nabla_U F(X_h, \mu)^\top \mid X_h]. \tag{29}$$

Since $U \perp X_h$, this becomes $\nabla_U F(X_h, \mu)\Sigma \nabla_U F(X_h, \mu)^\top$. Higher-order terms in the Taylor expansion of $F$ would contribute terms of $\mathcal{O}(\Sigma^2)$ etc.

For general nonlinear systems, the unconditional error covariance of the optimal model $f_{opt}(X_h)$ satisfies (using this first-order approximation for the conditional covariance):

$$\text{Cov}(\epsilon) \succeq \mathbb{E}_{X_h}\left[\nabla_U F(X_h, \mu)\Sigma \nabla_U F(X_h, \mu)^\top\right]. \tag{30}$$

This lower bound reflects the inherent system stochasticity due to $U$ and its propagation through the system dynamics $\nabla_U F$. $\qquad\square$

### B.1.4 JUSTIFICATION OF PROPOSITION B.1: UNIVERSALITY OF THE SELF-STIMULATION ERROR FLOOR

**Proposition B.1** (Self-Stimulation Error Floor). *For any self-stimulated model $\hat{X}_f = f(X_h)$ applied to a system $X_f = F(X_h, U)$ where $U \perp X_h$, $\mathbb{E}[U] = \mu$, $Cov(U) = \Sigma$, the error covariance $Cov(\epsilon) = \mathbb{E}[(\epsilon - \mathbb{E}[\epsilon])(\epsilon - \mathbb{E}[\epsilon])^\top]$ (or $\mathbb{E}[\epsilon\epsilon^\top]$ if $\mathbb{E}[\epsilon] = 0$) satisfies the following lower bound:*

$$Cov(\epsilon) \succeq \mathbb{E}_{X_h}\left[Cov_U(F(X_h, U) \mid X_h)\right]. \tag{31}$$

*Using a first-order approximation $Cov_U(F(X_h, U) \mid X_h) \approx \nabla_U F(X_h, \mu)\Sigma \nabla_U F(X_h, \mu)^\top$, this becomes:*

$$Cov(\epsilon) \succeq \mathbb{E}_{X_h}\left[\nabla_U F(X_h, \mu)\Sigma \nabla_U F(X_h, \mu)^\top\right]. \tag{32}$$

*Justification of Proposition B.1.* This proposition highlights that the self-stimulation error floor arises from fundamental system properties.

1. Intrinsic Limitation of Self-Stimulation: The error floor is fundamentally caused by the model's inability to account for the specific realization of the stochastic influence $U$, as it only has access to $X_h$. Even if the model $f(X_h)$ perfectly learns the true conditional mean behavior of the system, i.e., $f(X_h) = \mathbb{E}_U[F(X_h, U) \mid X_h]$, the inherent variability of $F(X_h, U)$ around this mean, $F(X_h, U) - \mathbb{E}_U[F(X_h, U) \mid X_h]$, introduces an irreducible noise component whose variance cannot be eliminated by any function of $X_h$ alone.

2. Generalization Across System Classes:

- **Linear Systems:** If $F(X_h, U) = AX_h + BU$, then $\mathbb{E}_U[F(X_h, U) \mid X_h] = AX_h + B\mu$. The irreducible error term is $B(U - \mu)$. Its covariance is $B\Sigma B^\top$. The gradient $\nabla_U F(X_h, \mu) = B$. The right-hand side of Eq. equation 32 becomes $\mathbb{E}_{X_h}[B\Sigma B^\top] = B\Sigma B^\top$, matching the exact result for linear systems.

- **Nonlinear Systems:** If $F(X_h, U)$ is nonlinear, the exact irreducible error covariance is $\mathbb{E}_{X_h}[\text{Cov}_U(F(X_h, U) \mid X_h)]$. The approximation $\mathbb{E}_{X_h}\left[\nabla_U F(X_h, \mu)\Sigma\nabla_U F(X_h, \mu)^\top\right]$ captures the first-order effect of $U$'s variance. The bound's magnitude depends on the structure of $F$, but a lower bound due to $U$ always holds.

3. Independence-Driven Irreducibility: The independence $U \perp X_h$ is crucial. It implies that $\mathbb{E}_U[F(X_h, U) \mid X_h] = \mathbb{E}_U[F(X_h, U)]$, and it ensures that the error covariance decomposes additively. Let $f_{opt}(X_h) = \mathbb{E}_U[F(X_h, U) \mid X_h]$. The total error is $\epsilon = (F(X_h, U) - f_{opt}(X_h)) + (f_{opt}(X_h) - f(X_h))$. The covariance $\text{Cov}(\epsilon)$ is the sum of the covariances of these two terms because the cross-term vanishes:

$$\mathbb{E}\left[(f_{opt}(X_h) - f(X_h))(F(X_h, U) - f_{opt}(X_h))^\top\right]$$

$$= \mathbb{E}_{X_h}\left[(f_{opt}(X_h) - f(X_h))\mathbb{E}_U[(F(X_h, U) - f_{opt}(X_h))^\top \mid X_h]\right] = 0,$$

since $\mathbb{E}_U[F(X_h, U) - f_{opt}(X_h) \mid X_h] = 0$ by definition of $f_{opt}$. The term $\text{Cov}(F(X_h, U) - f_{opt}(X_h))$ is the irreducible part.

**Implications of Proposition B.1:** This provides a definitive justification for the existence of an error floor:

- Self-stimulated models $f(X_h)$ are fundamentally constrained to predicting the conditional expectation $\mathbb{E}[X_f \mid X_h]$. They cannot predict the specific deviation from this mean caused by the unobserved realization of $U$.

- The error floor, characterized by $\mathbb{E}_{X_h}[\text{Cov}_U(F(X_h, U) \mid X_h)]$ (and approximated by Eq. equation 32), is fundamental. It arises from the system's inherent stochastic properties due to $U$ and its sensitivity to $U$, not from any particular choice of model $f(X_h)$ (assuming $f(X_h)$ can at best learn $\mathbb{E}[X_f \mid X_h]$).

- To reduce or eliminate this error floor, it is necessary to gain information about $U$, for example, by incorporating external measurements related to $U$ or by explicitly modeling $U$'s dynamics if possible, thus going beyond simple self-stimulation based on $X_h$ alone.

$\square$

## B.2 PROPOSITION 3.1: INFLUENCE EFFICACY

*Proof.* Consider a system with $p$ independent influences:

$$U_t = \sum_{i=1}^{p} U_i^t, \quad U_i^t \sim \mathcal{N}(\mu_i, \Sigma_i) \text{ (i.i.d.)}. \tag{33}$$

Let the true dynamics be $X_f = F(X_h, U_t)$, and let the model incorporate a subset of known influences $\{U_j^t\}$:

$$\hat{X}_f = f_\theta(X_h, U_j^t). \tag{34}$$

The prediction error is:

$$\epsilon = X_f - \hat{X}_f = F(X_h, U_t) - f_\theta(X_h, U_j^t). \tag{35}$$

Now, decompose $U_t$ into known ($U_j^t$) and unknown ($U_{-j}^t$) components:

$$\epsilon = \underbrace{F(X_h, U_j^t, U_{-j}^t) - F(X_h, U_j^t, \mu_{-j})}_{\text{Reducible Error}} + \underbrace{F(X_h, U_j^t, \mu_{-j}) - f_\theta(X_h, U_j^t)}_{\text{Model Mismatch}}, \tag{36}$$

where $\mu_{-j} = \mathbb{E}[U_{-j}^t]$.

Next, under optimal training, $f_\theta$ minimizes the mean squared error. This forces:

$$f_\theta^*(X_h, U_j^t) = \mathbb{E}_{U_{-j}^t}[F(X_h, U_j^t, U_{-j}^t) \mid X_h, U_j^t]. \tag{37}$$

The reducible error then simplifies to:

$$\epsilon = F(X_h, U_j^t, U_{-j}^t) - F(X_h, U_j^t, \mu_{-j}). \tag{38}$$

Now, consider the covariance reduction analysis:

**1. Linear Systems:** For $F(X_h, U_t) = AX_h + \sum_{i=1}^p B_i U_i^t$, the prediction error becomes:

$$\epsilon = \sum_{i \neq j} B_i(U_i^t - \mu_i). \tag{39}$$

The error covariance reduces by:

$$\Delta\text{Cov}(\epsilon) = B_j \Sigma_j B_j^\top. \tag{40}$$

**2. Nonlinear Systems:** For general $F(X_h, U_t)$, approximate via Taylor expansion at $U_{-j}^t = \mu_{-j}$:

$$\epsilon \approx \nabla_{U_{-j}} F(X_h, U_j^t, \mu_{-j})(U_{-j}^t - \mu_{-j}). \tag{41}$$

The covariance reduction becomes:

$$\Delta\text{Cov}(\epsilon) = \nabla_{U_j} F \Sigma_j (\nabla_{U_j} F)^\top. \tag{42}$$

Next, the independence argument:

The independence $U_j^t \perp U_{-j}^t$ ensures:

$$\text{Cov}(\epsilon) = \text{Cov}(\text{Reducible Error}) + \text{Cov}(\text{Model Mismatch}). \tag{43}$$

Optimal training nullifies the model mismatch term, leaving:

$$\text{Cov}(\epsilon) \succeq \sum_{i \neq j} \nabla_{U_i} F \Sigma_i (\nabla_{U_i} F)^\top. \tag{44}$$

This matches Proposition 3.1's claim.

Finally, we align with Proposition 2.1. The irreducible error floor in Proposition 2.1 is partially "carved out" by incorporating $U_j^t$. The reduction $\Delta\text{Cov}(\epsilon)$ quantifies how much influence knowledge lifts the theoretical performance ceiling.

This concludes the justification that Proposition 3.1 rigorously formalizes the intuition that *any measurable influence knowledge reduces forecasting uncertainty*, even under partial observability. $\square$

### B.2.1 CASE STUDY: DUAL-INFLUENCE LINEAR SYSTEM

**System Setup** Consider a linear system with two independent influences:

$$X_f = AX_h + B_1U_1 + B_2U_2, \quad U_1 \sim \mathcal{N}(0, \sigma_1^2), \ U_2 \sim \mathcal{N}(0, \sigma_2^2), \tag{45}$$

where:

$$A = \begin{bmatrix} 0.8 & 0 \\ 0 & 0.8 \end{bmatrix}, \ B_1 = \begin{bmatrix} 1 \\ 0 \end{bmatrix}, \ B_2 = \begin{bmatrix} 0 \\ 1 \end{bmatrix}, \ \sigma_1^2 = 0.5, \ \sigma_2^2 = 0.3. \tag{46}$$

The self-stimulated baseline model is:

$$\hat{X}_f^{(\text{base})} = CX_h + d. \tag{47}$$

**Case 1: No influence Knowledge** Using least squares, the optimal parameters are:

$$C^* = A, \quad d^* = B_1\mu_1 + B_2\mu_2 = 0 \quad (\text{since } \mu_1 = \mu_2 = 0). \tag{48}$$

Prediction error:

$$\epsilon^{(\text{base})} = B_1U_1 + B_2U_2. \tag{49}$$

Error covariance:

$$\text{Cov}(\epsilon^{(\text{base})}) = B_1\sigma_1^2 B_1^\top + B_2\sigma_2^2 B_2^\top = \begin{bmatrix} 0.5 & 0 \\ 0 & 0.3 \end{bmatrix}. \tag{50}$$

**Case 2: Partial influence Knowledge (Observing $U_1$)** Extend the model to leverage $U_1$:

$$\hat{X}_f^{(\text{IATSF})} = AX_h + B_1U_1 + d. \tag{51}$$

Optimal bias term:

$$d^* = B_2\mu_2 = 0. \tag{52}$$

Prediction error:

$$\epsilon^{(\text{IATSF})} = B_2(U_2 - \mu_2) = B_2U_2. \tag{53}$$

Error covariance:

$$\text{Cov}(\epsilon^{(\text{IATSF})}) = B_2\sigma_2^2 B_2^\top = \begin{bmatrix} 0 & 0 \\ 0 & 0.3 \end{bmatrix}. \tag{54}$$

The error is reduced by:

$$\Delta\text{Cov}(\epsilon) = \begin{bmatrix} 0.5 & 0 \\ 0 & 0 \end{bmatrix} = B_1\sigma_1^2 B_1^\top. \tag{55}$$

This matches Proposition 3.1 for linear systems.

**Case3: Nonlinear Extension** For a weakly nonlinear system $X_f = AX_h + \sin(U_1)B_1 + U_2B_2$:

• **Unknown $U_1, U_2$:**

$$\text{Cov}(\epsilon) \approx B_1\cos^2(\mu_1)\sigma_1^2 B_1^\top + B_2\sigma_2^2 B_2^\top = \begin{bmatrix} 0.5\cos^2(0) & 0 \\ 0 & 0.3 \end{bmatrix}. \tag{56}$$

• **Known $U_1$:**

$$\text{Cov}(\epsilon) \approx B_2\sigma_2^2 B_2^\top = \begin{bmatrix} 0 & 0 \\ 0 & 0.3 \end{bmatrix}. \tag{57}$$

**Conclusion**: Even in nonlinear systems, measurable influences reduce the error bound by their sensitivity-weighted variance, as formalized in Proposition 3.1.

## B.3   ERROR INTRODUCED BY INFLUENCE FORECASTER AND BENCHMARK DESIGN

### B.3.1   ERROR PROPAGATION WITH NON-OPTIMIZABLE INFLUENCE FORECASTING

Consider a linear system with historical state $X_h$ and future influence $U_f$:

$$X_f = AX_h + BU_f + w_h, \quad w_h \sim \mathcal{N}(0, \Sigma_w) \text{ (process noise)}, \tag{58}$$

where $U_f$ impacts the system instantaneously. Thus, - In training Phase: Uses true historical-future pairs $(X_h, X_f, U_f)$. - Testing Phase: Requires forecasting $U_f$ externally. The forecaster is *fixed* (not optimizable) and produces:

$$\hat{U}_f = U_f + \epsilon_f, \quad \epsilon_f \sim \mathcal{N}(0, \Sigma_{\hat{U}}). \tag{59}$$

After training, the model $\hat{X}_f = AX_h + BU_f$ achieves zero error if $\Sigma_w = 0$:

$$\epsilon_{\text{train}} = X_f - \hat{X}_f = w_h \quad \Rightarrow \quad \text{Cov}(\epsilon_{\text{train}}) = \Sigma_w. \tag{60}$$

In testing, predictions use the fixed forecaster $\hat{U}_f$:

$$\hat{X}_f = AX_h + B\hat{U}_f = AX_h + B(U_f + \epsilon_f). \tag{61}$$

The prediction error becomes:

$$\epsilon_{\text{test}} = X_f - \hat{X}_f = \underbrace{w_h}_{\text{System Noise}} - \underbrace{B\epsilon_f}_{\text{Irreducible Forecaster Error}}. \tag{62}$$

Error covariance (assuming $w_h \perp \epsilon_f$):

$$\text{Cov}(\epsilon_{\text{test}}) = \Sigma_w + B\Sigma_{\hat{U}}B^\top. \tag{63}$$

Thus, we find that, 1. The term $B\Sigma_{\hat{U}}B^\top$ dominates if $\Sigma_w \ll B\Sigma_{\hat{U}}B^\top$. This error is *independent of model quality*. 2. Since $\Sigma_{\hat{U}}$ is fixed and external, test error does not reflect the model's inherent capability. A "good" model may appear poor due to a low-quality forecaster.

CASE STUDY: SEPARATING MODEL AND FORECASTER EFFECTS

Let $A = I$, $B = I$, $\Sigma_w = 0$, and $\Sigma_{\hat{U}} = 0.5I$, we can train a perfect model: $\hat{X}_f = X_h + \hat{U}_f$.

But its test error gives:

$$\text{Cov}(\epsilon_{\text{test}}) = 0 + I \cdot 0.5I \cdot I^\top = 0.5I. \tag{64}$$

Despite a perfect model, test error is entirely dictated by $\Sigma_{\hat{U}}$.

**Final Conclusion**: When influences are forecasted by a non-optimizable external module, the test error upper bound is fundamentally constrained by:

$$\text{Cov}(\epsilon_{\text{test}}) \succeq B\Sigma_{\hat{U}}B^\top \tag{65}$$

This invalidates isolated model evaluation—performance metrics inherently conflate model and forecaster limitations.

### B.3.2 PERFECT INFLUENCE FORECASTER ASSUMPTION FOR FAIRNESS

To eradicate the noise introduced by the inaccurate influence forecaster for a fair benchmarking we assume that we have a perfect influence forecaster.

#### ASSUMPTION OF ACCURATE FORECASTER

Assume the influence forecaster is highly accurate, with negligible error:

$$\Sigma_{\hat{U}} \approx 0 \quad \Rightarrow \quad \hat{U}_f \approx U_f. \tag{66}$$

In this idealized scenario, the test-time prediction error reduces to:

$$\mathrm{Cov}(\epsilon_{\text{test}}) = \Sigma_w + B \cdot 0 \cdot B^\top = \Sigma_w. \tag{67}$$

#### IMPLICATIONS FOR MODEL EVALUATION

1. **Fair Assessment**: With $\Sigma_{\hat{U}} \approx 0$, the test error $\mathrm{Cov}(\epsilon_{\text{test}}) = \Sigma_w$ directly reflects the model's inherent capability, as it matches the training error bound.

2. **Decoupling Forecaster Effects**: A perfect forecaster eliminates the confounding term $B\Sigma_{\hat{U}}B^\top$, isolating the model's performance. This allows direct comparison between different models or training methodologies.

3. **Revealing True Limitations**: Any residual error $\Sigma_w$ now purely represents: - Fundamental system noise (unavoidable), - Model limitations (e.g., parameter estimation errors, structural mismatch).

### B.4 ERROR INTRODUCED BY WEIGHT SHARING

### B.4.1 LINEAR SYSTEM ANALYSIS

Consider a linear observation model with historical state $Z_h$ and multi-channel observations:

$$X_f = CZ_h = \begin{bmatrix} C_1 Z_h \\ C_2 Z_h \\ \vdots \\ C_k Z_h \end{bmatrix}, \quad C_i \in \mathbb{R}^{1 \times n}, \tag{68}$$

where $C_i$ is the distinct observation matrix for channel $i$.

Assume all channels share a single weight $c \in \mathbb{R}^{1 \times n}$:

$$\hat{X}_f = \mathbf{1}_k \cdot cZ_h = \begin{bmatrix} cZ_h \\ cZ_h \\ \vdots \\ cZ_h \end{bmatrix}. \tag{69}$$

The prediction error becomes:

$$\epsilon = X_f - \hat{X}_f = \begin{bmatrix} (C_1 - c)Z_h \\ (C_2 - c)Z_h \\ \vdots \\ (C_k - c)Z_h \end{bmatrix} = (C - \mathbf{1}_k c)Z_h. \tag{70}$$

Let $\Sigma_Z = \mathrm{Cov}(Z_h)$. The error covariance is:

$$\mathrm{Cov}(\epsilon) = (C - \mathbf{1}_k c)\Sigma_Z (C - \mathbf{1}_k c)^\top. \tag{71}$$

The optimal shared weight $c_{\text{opt}}$ minimizing the trace is:

$$c_{\text{opt}} = \frac{1}{k} \sum_{i=1}^{k} C_i.\tag{72}$$

Substituting $c_{\text{opt}}$, the irreducible error covariance becomes:

$$\text{Cov}(\epsilon_{\text{opt}}) = \left( C - \frac{1}{k}\mathbf{1}_k\mathbf{1}_k^\top C \right) \Sigma_Z \left( C - \frac{1}{k}\mathbf{1}_k\mathbf{1}_k^\top C \right)^\top.\tag{73}$$

### B.4.2 NONLINEAR SYSTEM GENERALIZATION

For nonlinear observations $X_f = \mathcal{O}(Z_h) = [o_1(Z_h), \ldots, o_k(Z_h)]^\top$, a weight-shared model forces:

$$\hat{X}_f = \mathbf{1}_k \cdot o(Z_h).\tag{74}$$

The error is:

$$\epsilon = \begin{bmatrix} o_1(Z_h) - o(Z_h) \\ \vdots \\ o_k(Z_h) - o(Z_h) \end{bmatrix}.\tag{75}$$

Assume $o_i(Z_h) = o(Z_h) + \Delta_i(Z_h)$ with $\Delta_i \sim \mathcal{N}(0, \Sigma_i)$. The covariance becomes:

$$\text{Cov}(\epsilon) = \text{diag}(\Sigma_1, \ldots, \Sigma_k).\tag{76}$$

### B.4.3 JUSTIFICATION: KEY ANALOGIES TO PREVIOUS FRAMEWORK

- **Structural Bias**: Weight-sharing corresponds to assuming $U_f$ is constant across channels, analogous to ignoring external influences.

- **Irreducible Error**: The term $\text{Cov}(\epsilon_{\text{opt}})$ mirrors $B\Sigma B^\top$, where $\Sigma$ represents unmodeled channel-specific variations.

- **Sensitivity Amplification**: The matrix $C - \mathbf{1}_k c_{\text{opt}}$ amplifies discrepancies, similar to $\nabla_U F$ in nonlinear systems.

### B.4.4 CASE STUDY: TWO CHANNELS

Let $o_1(Z_h) = Z_h$, $o_2(Z_h) = 2Z_h$, and force $o(Z_h) = aZ_h$. The optimal $a = 1.5$ yields:

$$\text{Cov}(\epsilon) = \frac{1}{4}\text{Var}(Z_h) \begin{bmatrix} 1 & 1 \\ 1 & 1 \end{bmatrix}.\tag{77}$$

**Conclusion**: Weight-sharing introduces an error floor governed by:

$$\text{Cov}(\epsilon) \succeq \left( C - \frac{1}{k}\mathbf{1}_k\mathbf{1}_k^\top C \right) \Sigma_Z \left( C - \frac{1}{k}\mathbf{1}_k\mathbf{1}_k^\top C \right)^\top \tag{78}$$

This matches the structure of Proposition 2.1, where unmodeled channel diversity plays the role of unobserved influences. Breaking this bound requires abandoning weight-sharing or introducing channel-specific adapters.

## B.5 ERROR INTRODUCED BY INCOMPLETE OBSERVATION

Finally, we would like to discuss the error introduced by incomplete observation which is also an inherent error source in the TSF. This shows that our given lower bound is already a very ideal and conservative, there is still a lot loophole in the TSF task formulation.

Consider a hidden state system with partial observations:

$$Z_f = AZ_h + BU, \quad U \sim \mathcal{N}(\mu, \Sigma), \tag{79}$$

where $Z_h \in \mathbb{R}^n$ is the historical hidden state. The observable state is:

$$X_h = HZ_h, \quad H \in \mathbb{R}^{m \times n}, \text{ rank}(H) = m < n. \tag{80}$$

The observable dynamics become:

$$X_f = HZ_f = HAZ_h + HBU. \tag{81}$$

The hidden state can be decomposed as:

$$Z_h = H^+ X_h + \tilde{Z}_h, \tag{82}$$

where $H^+$ is the pseudo-inverse of $H$, and $\tilde{Z}_h$ represents the unobservable state component. Substituting into $X_f$:

$$X_f = HAH^+ X_h + HA\tilde{Z}_h + HBU. \tag{83}$$

A self-stimulated model predicts:

$$\hat{X}_f = CX_h + d. \tag{84}$$

The prediction error is:

$$\epsilon = X_f - \hat{X}_f = (HAH^+ - C)X_h + HA\tilde{Z}_h + HB(U - \mu). \tag{85}$$

The least squares solution gives:

$$C^* = HAH^+, \quad d^* = HB\mu. \tag{86}$$

The irreducible error becomes:

$$\epsilon = HA\tilde{Z}_h + HB(U - \mu). \tag{87}$$

The error covariance splits into two components:

$$\text{Cov}(\epsilon) = \underbrace{HA\text{Cov}(\tilde{Z}_h)(HA)^\top}_{\text{Hidden State Error}} + \underbrace{HB\Sigma B^\top H^\top}_{\text{influence Error}}. \tag{88}$$

Finally, the error lower bound comes from: 1. **Hidden State Error**: Propagates through $HA$ from the unobservable subspace, governed by $\text{Cov}(\tilde{Z}_h)$. 2. **influence Error**: Matches Proposition 2.1's bound $B\Sigma B^\top$, projected onto the observable space via $H$.

The total error lower bound becomes:

$$\text{Cov}(\epsilon) \succeq HB\Sigma B^\top H^\top + HA\text{Cov}(\tilde{Z}_h)(HA)^\top \tag{89}$$

This extends Proposition 2.1 by adding a term from partial observability. The bound is conservative because:

- Hidden state error $\text{Cov}(\tilde{Z}_h)$ depends on system stability in the unobservable subspace.
- Noisy influences $U$ remain irreducible without direct measurement.

# C RELATED WORKS

## C.1 TEXT EMBEDDING MODEL

Text embedding models have undergone significant advancements, providing efficient and semantically rich vector representations of textual information. Early transformer-based models like BERT (Devlin et al., 2019) encode sentences into embeddings by pretraining on masked language modeling tasks, enabling them to capture contextual semantics. However, BERT embeddings are not specifically optimized for tasks requiring fine-grained semantic similarity, prompting the development of more task-specific models.

MPNet (Song et al., 2020) and MiniLM (Wang et al., 2020) build upon BERT () by introducing novel architectural and pretraining strategies. MPNet combines masked language modeling with permuted sequence prediction, allowing for better contextual understanding and token dependencies. MiniLM, on the other hand, employs knowledge distillation to create smaller, faster models that retain high performance, making them ideal for resource-constrained applications.

OpenAI's embedding models (ope) represent another major step forward, leveraging large-scale proprietary transformer architectures. These embeddings are designed to excel in tasks like semantic search, classification, and similarity, offering generalizability and strong performance across a variety of applications. They also incorporate dimensional flexibility, allowing embeddings to be truncated or adjusted based on application needs, as seen with the Matryoshka embedding technique. This technique allows embeddings to maintain their semantic integrity even when their dimensions are reduced, offering scalability and adaptability.

A key property of text embeddings is their compatibility with similarity measures like cosine similarity. By projecting text into a shared semantic space, cosine similarity enables the computation of semantic closeness between embeddings, making it a foundational operation for tasks like clustering, retrieval, and alignment between modalities. This capability is crucial in applications requiring robust generalization across diverse textual expressions.

Together, these advancements have expanded the utility of text embeddings in various domains, including information retrieval, natural language understanding, and multimodal learning tasks. Our work builds on these innovations by leveraging pre-trained text embeddings for aligning textual semantics with time series patterns, ensuring robust causal modeling and efficient text-guided time series forecasting.

## C.2 TIME SERIES ANALYSIS WITH TEXT EMBEDDING

Adding more information to time series by incorporating heterogeneous information has been a long-studied topic, with several works opting to use text embeddings as input.

In the financial field, where time series are often more correlated to external information, several works (Sawhney et al., 2021; Liu et al., 2024b) have used text embeddings as external graph relationships to capture the correlations between keywords and stock descriptions, further influencing the ranking process in stock trading. More recently, a line of works (Liu et al., 2024a; Jia et al., 2024) has sought to enrich time series data by adding news text embeddings to the time series embeddings. However, these methods still face limitations in solving information insufficiency, as they do not incorporate causal information that could guide the model in predicting time series patterns driven by external events. Additionally, these works primarily use external text embeddings to expand the lookback window, without fully exploiting the underlying properties of the text embeddings.

To tackle these challenges, we introduce the Time-Series Guided Text Forecasting (IATSF) model, which expands traditional time series forecasting by incorporating external textual data that offers causal insights. Unlike previous approaches that use text embeddings simply as supplementary information, IATSF leverages the text to provide causal guidance, aligning textual data with time series patterns. Through the integration of CASM, we can effectively extract channel-dynamic news

correlations from the pre-trained text embeddings, enabling the model to adapt to the specific distributions of different time series channels. This allows the model to make more accurate predictions by incorporating both the semantic meaning of the text and its causal relationship with the time series data.

## D  ABOUT PREDICTABILITY OF TREND

In our study, we define "**trend**" as patterns that exhibit very low frequency while lacking periodicity within the observed time window, rather than simple exponential or linear patterns. For instance, the pressure channel in our Atmospheric Physics dataset exemplifies this with its irregular low-frequency fluctuations, which appear to be random and non-periodic. Such randomness hampers the model's ability to learn stable patterns when relying solely on historical time series data.

However, these low-frequency patterns often correlate with external influences—for example, a drop in temperature due to cold air can significantly increase atmospheric pressure. By integrating this type of external information, our model is designed to discern causal relationships between such environmental factors and the observed low-frequency trends, thereby enhancing predictability.

For a practical illustration, please refer to the pressure (p-bar) channel in Fig. 3 and Fig. 7. This channel displays non-periodic fluctuations, which the traditional patchTST model even struggles produce a valid forecasting. In contrast, our IATSF model, which incorporates external textual cues, successfully tracks these changes, demonstrating the effectiveness of including external information for predicting complex trends.

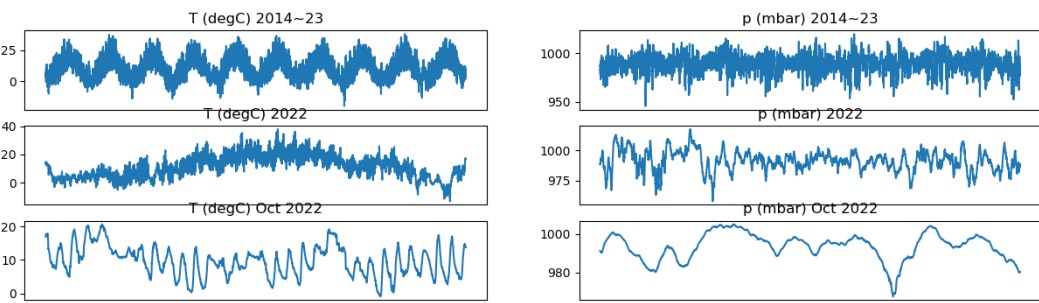

Figure 7: Visualization on two channels with different time scale. Temperature channel shows obvious periodicity both daily and annually. However the atmosphere pressure seems to be noise on large time scale but shows slowly random changing low-frequency "trend". Which makes it hard to be predicted without external information.

## E  EXPERIMENT SETTINGS

**Toy Dataset** For Toy dataset, All of the models are following the same experimental setup with prediction length $H \in \{14, 28, 60, 120\}$ and LBW length $T = 60$.

**Electrical Utility** For Electrical Utility dataset we follow the experiment settings in previous works as follows: forecasting horizon $H \in \{96, 192, 336, 720\}$, look back window length of 288. For fair comparison, we directly compare with the results report in the baseline original paper. And the FIATS is trained with the captioned version. On this dataset, we also test the impact of a shorter look-back window on the FIATS.

**Atmospheric Physic** We follow the experiment settings in previous works as follows: forecasting horizon $H \in \{96, 192, 336, 720\}$, look back window length of 360. We trained all other models on the Atmospheric Physic 14-19 and 14-24 dataset with their setting on the original weather dataset accordingly. And the FIATS is trained with the captioned version.

**GAUD** We split the dataset on each time series by 7:1:2 for training, validating and testing. We set the forecasting horizon H for 14 and look back window length of 60. For pretraining, we concatenate all the training set together to train and validate the model and test on each test set separately.

## F  CHANNEL-WISE PERFORMANCE ON ATMOSPHERIC PHYSICS DATASET

The difficulty in predicting each channel varies, therefore, we present channel-wise performance in Table 5. The results demonstrate that FIATS, with the aid of external textual climate reports, significantly enhances forecasting accuracy across all channels. Notably, the model achieves over a 60% performance improvement in channels such as atmospheric pressure (p (mbar)), relative humidity (rh(%)), and vapor pressure deficit (VPdef (mbar)), which typically cannot be predicted reliably using historical time series data alone. The integration of external text cues has led to groundbreaking improvements in forecasting these parameters.

However, the wind velocity channel shows minimal variance in performance across all models, each achieving similar results with slight losses. This phenomenon is attributed to the presence of extreme values in this channel, which, after normalization, diminish the impact of more typical values on the overall gradient. Consequently, all models struggle to learn detailed patterns in this channel due to the reduced contribution to the global gradient.

Another noteworthy observation is that while FIATS is capable of predicting rainfall—unlike models that default to predicting near-zero averages—the performance improvement in these channels is modest. This is because rainfall is relatively scarce in this dataset, leading to large losses when rain is inaccurately predicted at the wrong times. Conversely, predicting the average value results in a smaller overall loss. This tendency explains why other models often opt for the average, avoiding the complex task of learning rainfall patterns. Nevertheless, accurate rainfall forecasting remains crucial in meteorological applications, underscoring our commitment to enhancing predictive accuracy in this area. The same principle also applies to other channels.

Table 5: Channel wise performance on Weather-medium dataset in MSE. The best is highlighted in bold and the second best is highlighted in underline.

| Channel | FIATS | FITS | DLinear | PatchTST | iTransformer | IMP. |
|---|---|---|---|---|---|---|
| p (mbar) | **0.1365** | 0.8637 | 0.8238 | 0.9301 | 1.0320 | **83.43%** |
| T (degC) | **0.1889** | 0.2924 | 0.3329 | 0.2964 | 0.3233 | 35.40% |
| Tpot (K) | **0.1829** | 0.3163 | 0.3525 | 0.3225 | 0.3533 | 42.18% |
| Tdew (degC) | **0.3467** | 0.4043 | 0.4085 | 0.4082 | 0.4258 | 14.25% |
| rh (%) | **0.2479** | 0.6541 | 0.6788 | 0.6997 | 0.8185 | **62.10%** |
| VPmax (mbar) | **0.2369** | 0.3500 | 0.3984 | 0.3521 | 0.4086 | 32.31% |
| VPact (mbar) | **0.2998** | 0.3404 | 0.3534 | 0.3515 | 0.3845 | 11.93% |
| VPdef (mbar) | **0.2835** | 0.6384 | 0.6968 | 0.6744 | 0.8038 | **55.59%** |
| sh (g/kg) | **0.2995** | 0.3434 | 0.3562 | 0.3557 | 0.3896 | 12.78% |
| H2OC (mmol/mol) | **0.2996** | 0.3434 | 0.3562 | 0.3556 | 0.3894 | 12.75% |
| rho (g/m³) | **0.1926** | 0.3909 | 0.4119 | 0.4182 | 0.4535 | **50.73%** |
| wv (m/s) | **0.0002** | **0.0002** | **0.0002** | **0.0002** | 0.0003 | 0.00% |
| max. wv (m/s) | **0.0004** | 0.0005 | **0.0004** | 0.0005 | 0.0006 | 0.00% |
| wd (deg) | **0.7270** | 1.1605 | 1.1295 | 1.1344 | 1.2735 | 35.64% |
| rain (mm) | **0.6824** | 0.6905 | 0.7167 | 0.6891 | 0.6998 | 0.97% |
| raining (s) | **0.7900** | 0.8735 | 0.9379 | 0.8591 | 0.9942 | 8.04% |
| SWDR (W/m²) | **0.1828** | 0.3084 | 0.3856 | 0.2967 | 0.3776 | 38.39% |
| PAR (umol/m²/s) | **0.1773** | 0.2840 | 0.3588 | 0.2704 | 0.3473 | 34.43% |
| max. PAR (umol/m²/s) | **0.1975** | 0.2599 | 0.3195 | 0.2632 | 0.3226 | 24.01% |
| Tlog (degC) | **0.1774** | 0.2802 | 0.3260 | 0.2806 | 0.3290 | 36.69% |
| CO2 (ppm) | **0.2600** | 0.2716 | 0.2812 | 0.2618 | 0.2760 | 0.69% |
| Avg. Loss | **0.2814** | 0.4317 | 0.4583 | 0.4391 | 0.4954 | 34.82% |

## G  PERFORMANCE VISUALIZATION ON ATMOSPHERIC PHYSICS DATASET

We provide the full visualization as Fig. 8. The FIATS shows great performance across all the channels. Even very hard ones such as Wind dir. It can also model the time series that totally independent with the weather such as the CO2 channel.

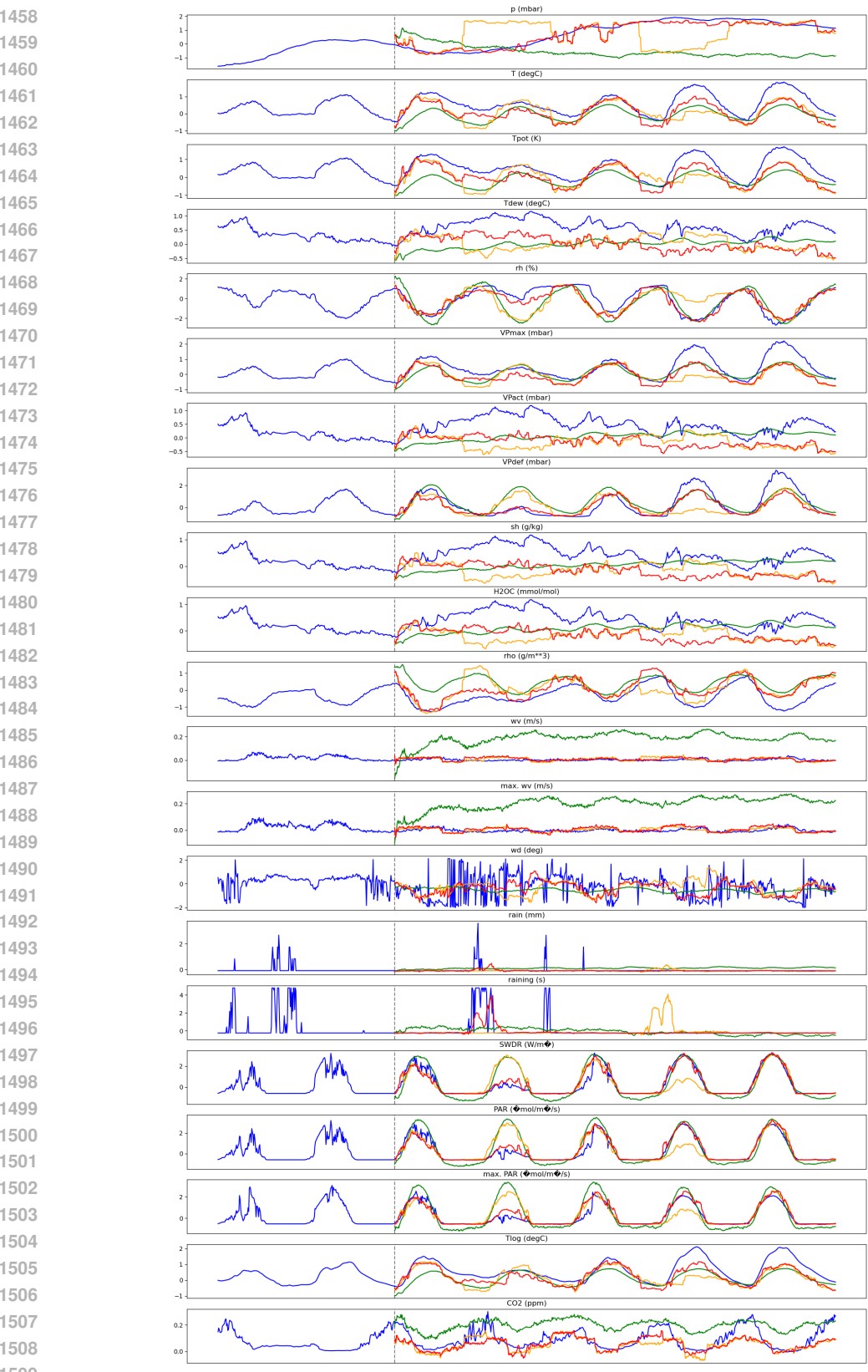

Figure 8: Full visualization all channels on the 15000th test sample of Weather-Caption-Medium dataset. Blue line for ground truth, Red line for FIATS, Green line for PatchTST and Orange line for FIATS with swapping the news on the second and forth forecasting day.

## H  WEATHER RESULTS W. W/O. RIN

We compared the performance of models with and without Reversible Instance Normalization (RIN) on the Atmospheric Physics-medium dataset, focusing on a 720-hour forecasting horizon. The model with RIN enabled achieved an MSE of 0.3428, whereas the model without RIN achieved a lower MSE of 0.2814. Results visualized in Fig. 9 show that the RIN-enabled model exhibits significant biases in many channels, particularly those with gradual trend shifts. This occurs because RIN removes the bias term from all instances, leaving the model unable to recognize relative bias and trend values. For instance, with RIN, temperature patterns in winter and summer are treated similarly, ignoring the typically higher and more variable temperatures in summer. Additionally, we noted pronounced shifting behavior coinciding with changes in captions, suggesting that the absence of bias information leads the model to over-rely on textual prompts, compensating for the missing data.

## I  ABLATION STUDY ON CAUSAL RELATIONSHIP EXTRACTION

FIATS is designed to learn causal relationships between events described in text and their corresponding time series patterns. While not explicitly an alignment model, it effectively aligns the semantic meaning of text with the time series data it impacts. The model generates time series patterns guided by the textual information, and its performance varies based on the quality of the text input:

1. **Training with Meaningful and Relevant Text:**

   - **Inference with Similar Text:** Produces strong results by accurately extracting causal relationships between events in the text and time series patterns.

2. **Training with Zero/Random Text:**

   - **Inference with Any Text:** Produces results equivalent to PatchTST, as no additional information is present in the text. The model relies solely on the time series data, ignoring the random text.

3. **Training with Meaningful Text, Inference with Incorrect Text:**

   - **Inference with Incorrect Text:** Results are poor, as the model relies on the misleading text input and generates patterns based on incorrect or irrelevant information.

We detail the FIATS performance under different text conditions in the following table:

Table 6: FIATS performance under different training and testing conditions. We report the result of forecasting horizon 96 on Atmospheric Physics-medium dataset using MiniLM embedding.

| | | Train with | | |
|---|---|---|---|---|
| | | Good | Zero | Random |
| Test with | Good | 0.186 (captures causal relationships) | 0.249 (corrupted random patterns) | 0.251 (similar to PatchTST) |
| | Zero | 0.724 (corrupted repetive patterns) | 0.249 (similar to PatchTST) | 0.254 (similar to PatchTST) |
| | Random | 0.615 (corrupted random patterns) | 0.249 (similar to PatchTST) | 0.250 (similar to PatchTST) |

The results of the ablation study provide strong evidence that FIATS relies on capturing causal relationships between time series patterns and dynamic news, rather than simply treating text as auxiliary input. When trained with meaningful and correlated news, the model demonstrates its ability to effectively extract these relationships, yielding strong predictive performance. This highlights FIATS's capacity to align the semantic meaning of text with time series patterns in a causally meaningful way.

On the other hand, when trained with good text but tested with random or misleading text, the model produces poor predictions because it continues to rely on the input text, even when it is inaccurate or irrelevant. This further underscores the model's dependence on the quality of the textual input rather than merely defaulting to learned time series patterns.

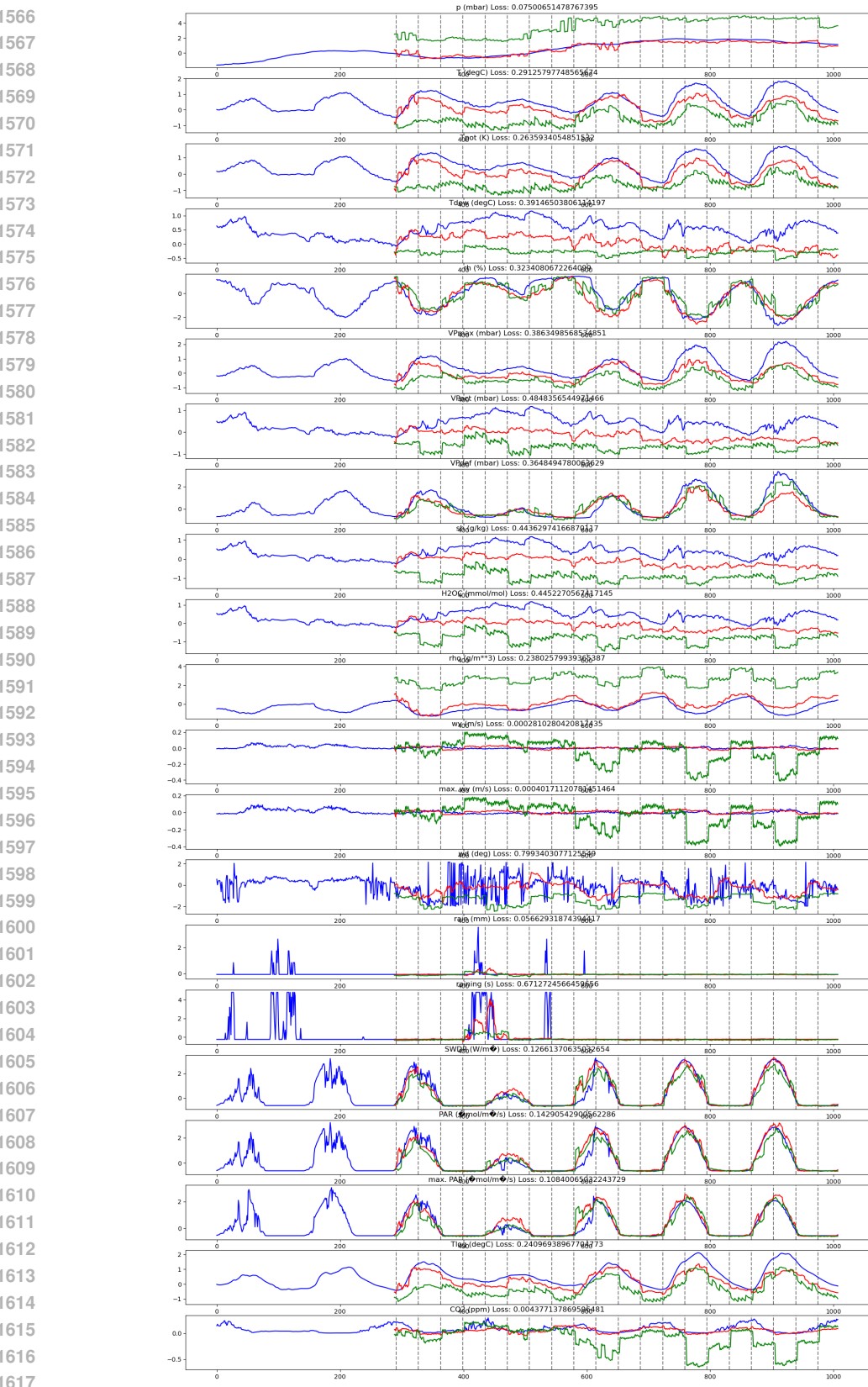

Figure 9: Full visualization all channels on the 15000th test sample of Weather-Caption-Medium dataset. Blue line for ground truth, Red line for FIATS without RIN, Green line for FIATS with RIN enabled.

Interestingly, when trained with bad or random text, FIATS fails to establish causal relationships and instead reverts to PatchTST-level performance, indicating it falls back to relying solely on time series data. Furthermore, when subsequently tested with good text, the model trained on bad text still ignores the input entirely, suggesting it stops depending on textual input when the training data lacks meaningful causal relationships.

These results collectively demonstrate that FIATS's strength lies in its ability to extract and leverage causal relationships between text and time series data. The model's performance is tightly coupled with the quality and relevance of the textual input, validating the centrality of causal alignment in its design and functionality.

These outcomes demonstrate that IATSF effectively achieves alignment in the "event" space, linking events described in the text to the corresponding time series patterns.

## J    ATTENTION MAP VISUALIZATION ON ATMOSPHERIC PHYSICS

We further visualize two cross-attention blocks to further investigate the FIATS. You are strongly advised to check the Tab. 13, Appendix O.4.5 and Fig. 8 while reading this part.

Figure 10 illustrates the attention map of the "text-guided channel independent" cross-attention block in the text encoder across three layers. In the first layer, attention is predominantly focused on the first sentence, which specifies the month and time. This sentence is crucial as it provides temporal context that significantly impacts the prediction of both daily and annual periodicity. While other sentences receive moderate attention, the sixth sentence, which describes atmospheric pressure as detailed in Table 13, consistently receives no attention across all channels.

In the second layer, however, there is a notable shift in attention dynamics. All channels, particularly channel 0, show intense focus on the sixth sentence. According to the channel definitions in Appendix O.4.5, channel 0 directly corresponds to atmospheric pressure. Channels 10 and 20, which are related to air density and $CO_2$ concentration respectively—factors closely associated with pressure—also display relatively high attention scores. This suggests that the FIATS is capable of discerning the underlying relationships among the channels.

The separation of attention focus between the first and second layers suggests that the influence of atmospheric pressure on the model's predictions is independent of time. In the third layer, a diversity of attention patterns emerges; channel 0 focuses exclusively on the sixth sentence, while other channels predominantly attend to the first sentence.

Since we take the output of previous layer as query and input news embeddings as key and value, the information lies in the news are progressively added to the channel embeddings. Thus, the model can focus on different perspective in separate cross attention layers.

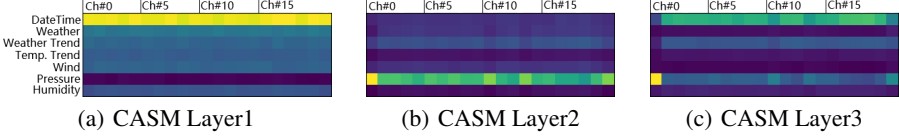

| (a) CASM Layer1 | (b) CASM Layer2 | (c) CASM Layer3 |

Figure 10: Attention map of the "CASM" cross attention block on the 15000th test sample of Atmospheric Physics dataset dataset. We use three cross attention block. The vertical axis stand for channels and horizon stand for the 7 sentences of the weather report summary.

Figure 11 presents the attention map of the modality mixer layer cross attention block in the Atmospheric Physics dataset. The map, averaged across three cross attention layers, illustrates distinct patterns of attention for each channel. This diversity underscores the FIATS's ability to adaptively extract time series embeddings tailored to the unique distribution characteristics of each channel, facilitated by textual inputs.

Notably, the channels for SWDR, PAR, and max.PAR display clear periodic patterns in their attention maps, aligning with observations from waveform visualizations. These patterns suggest that the FIATS effectively captures and utilizes periodic information from these environmental variables.

Furthermore, the channels labeled rain and raining show a particularly interesting behavior; they assign significantly higher attention scores to the exact time periods of rainfall within the look-back window. This behavior indicates that the FIATS is adept at identifying and prioritizing crucial

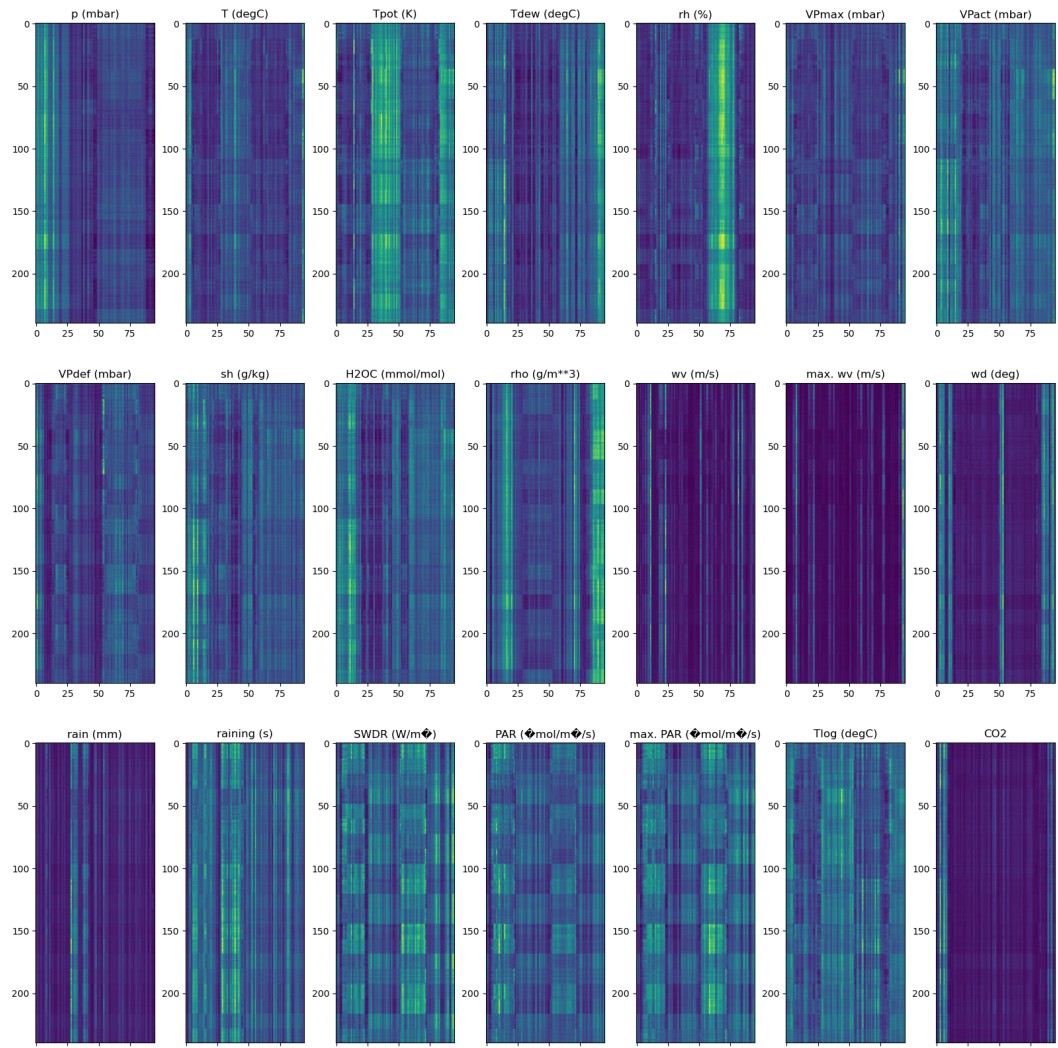

Figure 11: Attention map of the modality mixer layer cross attention block on Atmospheric Physics dataset, on the 15000th test sample of Weather-Caption-Medium dataset. The attention map is averaged across three cross attention layers. We plot the attention map for each channel. The vertical axis stand for output time series patches and the horizon stand for the input time series patches embedding from PatchTST backbone.

temporal events specific to each channel, further enhancing its forecasting accuracy by focusing on relevant patterns where needed. This level of detail in attention allocation demonstrates the model's capability to integrate contextual cues from textual data and further guide the time series forecasting.

## K   COMPARISON WITH MORE BASELINES ON ELECTRICAL UTILITY

We further compare with more baselines on Electrical Utility, including Autoformer, Fedformer, Informer, FiLM and TimesNet (Wu et al., 2021; Zhou et al., 2022a; 2021; 2022b; Wu et al., 2023). FIATS shows dominant superior performance across these baselines, as shown in Tab. 7.

Table 7: The comparison on Electricity dataset with other baselines. Best is marked in bold and the second best is marked in underline.

|  | Pred. Len. | FIATS | FIATS_120 | Autoformer | Fedformer | Informer | FiLM | TimesNet |
|---|---|---|---|---|---|---|---|---|
| Elec | 96 | **0.124** | 0.127 | 0.201 | 0.188 | 0.274 | 0.154 | 0.168 |
|  | 192 | **0.144** | 0.146 | 0.222 | 0.197 | 0.296 | 0.164 | 0.184 |
|  | 336 | **0.16** | 0.164 | 0.231 | 0.212 | 0.3 | 0.188 | 0.198 |
|  | 720 | **0.193** | 0.200 | 0.254 | 0.244 | 0.373 | 0.236 | 0.22 |

## L   FULL RESULT ON GAUD DATASET

We show the comparison on GAUD Dataset in Tab. 8, and Tab, 9 with PatchTST. We use de-normed MAE as metric since the base volume of players of each game varies drastically, using normed metrics can lead to unfair comparison. The pretrain indicate that the model is jointly trained on all the games and each game is labeled by the channel discription. The Gnorm indicate that we apply the global normalization to preserve the player variation mentioned before. But it seems bring limited boost.

Table 8: Full result on GAUD dataset in de-normalized MAE. The best result is shown in green shaded bold font. The ones with performance boost over 10% is marked in red.

| game_id | IATSF_pretrain | IATSF_pretrain_Gnorm | IATSF | PatchTST | IMP/% |
|---|---|---|---|---|---|
| 10 | 781.2257 | **724.5196** | 849.8968506 | 804.757019 | 0.099704 |
| 240 | **372.887** | 374.03445 | 400.0899353 | 421.2349243 | 0.114777 |
| 440 | **10216.276** | 10816.979 | 10694.47852 | 10828.37891 | 0.056528 |
| 550 | **4110.673** | 4437.846 | 4336.002441 | 4587.186035 | 0.103879 |
| 570 | **30179.111** | 30914.955 | 31916.02344 | 32031.38477 | 0.057827 |
| 620 | **674.1199** | 789.6992 | 799.2894897 | 710.8320313 | 0.051647 |
| 730 | 51687.87 | **50937.348** | 52638.79297 | 51069.08984 | 0.00258 |
| 3590 | **687.4657** | 775.8215 | 734.3128662 | 743.4817505 | 0.075343 |
| 39210 | **3310.422** | 3719.8003 | 3388.064453 | 3420.974609 | 0.032316 |
| 105600 | **4415.245** | 4899.586 | 4464.801758 | 4513.034668 | 0.021668 |
| 107410 | 1597.8743 | 1548.8544 | 1411.974243 | **1355.757813** | 0 |
| 214950 | 331.97876 | 350.3743 | 328.4424744 | **324.0109863** | 0 |
| 218620 | **5576.539** | 5714.511 | 5650.027832 | 5818.570313 | 0.041596 |
| 221100 | **2574.3164** | 2713.0063 | 3260.266113 | 2742.404053 | 0.061292 |
| 222880 | **50.29491** | 138.43388 | 61.51412582 | 59.99531174 | 0.161686 |
| 227300 | **3224.443** | 3356.0312 | 3418.429199 | 3388.108398 | 0.048306 |
| 230410 | **5307.4634** | 5733.2676 | 5856.409668 | 6040.648926 | 0.121375 |
| 231430 | 441.79767 | **377.2723** | 581.7993774 | 713.0888062 | 0.470932 |
| 232050 | **5.8684874** | 140.72949 | 6.452753067 | 6.638870239 | 0.116041 |
| 236390 | **4528.812** | 4847.2886 | 4787.857422 | 4680.506348 | 0.03241 |
| 236850 | 1197.9114 | 1308.7864 | **1169.570313** | 1191.723145 | 0.018589 |
| 242760 | **5888.873** | 6968.5566 | 6002.225586 | 6160.327148 | 0.044065 |
| 244210 | **657.4606** | 672.32324 | 676.5147095 | 658.0761108 | 0.000935 |
| 250900 | 895.2101 | **886.4052** | 1012.618652 | 892.572937 | 0.00691 |
| 251570 | 4304.9175 | 4886.079 | **4088.169922** | 4352.344727 | 0.060697 |
| 252950 | 1971.4385 | **1928.533** | 1971.546509 | 2054.135742 | 0.061146 |
| 255710 | 2154.8572 | 2082.0603 | 2231.175781 | **2078.98584** | 0 |
| 270880 | 789.74634 | 748.85596 | **746.2268677** | 760.31073 | 0.018524 |
| 271590 | **9292.364** | 9546.938 | 10758.82422 | 9438.995117 | 0.015535 |
| 275850 | **2671.1484** | 3121.9731 | 3179.639404 | 3118.741699 | 0.143517 |
| 281990 | **2094.3948** | 2404.5767 | 3269.309326 | 2315.452881 | 0.095471 |
| 284160 | 929.92413 | **903.6123** | 956.4987183 | 908.8114014 | 0.005721 |
| 289070 | 4861.033 | 5243.972 | 4706.715332 | **4633.394531** | 0 |
| 291550 | 1339.3384 | 1323.4893 | **1290.662231** | 1324.343018 | 0.025432 |
| 292030 | **3181.2307** | 3723.4417 | 3607.125732 | 3500.928223 | 0.091318 |
| 294100 | 2123.2292 | 2150.6 | **1990.630981** | 2074.705322 | 0.040524 |
| 304930 | 5698.3926 | 5597.64 | **5430.058105** | 5824.242188 | 0.06768 |
| 306130 | **1727.7567** | 1927.9446 | 1787.971802 | 1765.812744 | 0.021552 |
| 322170 | 987.1338 | **878.4059** | 902.7176514 | 910.0273438 | 0.034748 |
| 322330 | **4585.129** | 5155.708 | 4900.762207 | 4916.681152 | 0.067434 |
| 346110 | **6389.454** | 7178.1084 | 7004.43457 | 6871.126953 | 0.070101 |
| 359550 | 5251.984 | 5247.6694 | 5184.080566 | **5156.916016** | 0 |
| 364360 | **78.73614** | 182.35359 | 89.08701324 | 88.43521118 | 0.109674 |
| 365590 | **158.19469** | 229.93842 | 173.8761902 | 180.6734314 | 0.124416 |
| 374320 | **557.42993** | 630.0981 | 659.1234131 | 655.4638672 | 0.149564 |
| 377160 | 1733.9108 | **1486.6573** | 1814.430298 | 1725.577515 | 0.138458 |
| 381210 | **5498.251** | 5568.4336 | 5691.748047 | 5964.625488 | 0.07819 |
| 386360 | 1135.4182 | 1231.7946 | 1126.588989 | **1125.822021** | 0 |
| 394360 | 2788.8633 | **2674.6377** | 2861.023682 | 2725.428223 | 0.018636 |
| 413150 | 3061.9893 | 3204.8018 | 2831.16748 | **2713.414551** | 0 |

Table 9: Cont. Full result on GAUD dataset in de-normalized MAE. The best result is shown in green shaded bold font. The ones with performance boost over 10% is marked in red.

| game_id | IATSF_pretrain | IATSF_pretrain_Gnorm | IATSF | PatchTST | IMP/% |
|---|---|---|---|---|---|
| 427520 | 923.96985 | **772.3062** | 803.0150146 | 775.4301758 | 0.004029 |
| 457140 | 1159.0262 | 1217.2543 | 1133.615601 | **1107.060547** | 0 |
| 489830 | 2071.2698 | 2067.1377 | 1855.288574 | **1809.03479** | 0 |
| 493520 | **270.12488** | 337.2064 | 311.7146606 | 314.7081604 | 0.141665 |
| 513710 | **1662.712** | 1732.691 | 2458.639648 | 2096.899658 | 0.207062 |
| 526870 | 1714.113 | **1704.2557** | 2002.178223 | 2031.111084 | 0.160924 |
| 529340 | **2037.8783** | 3383.1628 | 4038.390381 | 3701.234863 | 0.449406 |
| 548430 | **3215.861** | 3539.4717 | 3559.275391 | 3622.380615 | 0.112224 |
| 552500 | **2653.9133** | 2886.299 | 3560.82251 | 4062.790527 | 0.346776 |
| 552990 | 5025.465 | 5687.0366 | **3085.568848** | 5004.588379 | 0.383452 |
| 578080 | 21942.736 | **20576.87** | 24925.38086 | 21725.19727 | 0.052857 |
| 582010 | **2820.5447** | 3062.556 | 3161.88623 | 3088.219727 | 0.086676 |
| 582660 | **1543.2185** | 1686.1365 | 1597.595703 | 1613.270874 | 0.043423 |
| 646570 | **1219.2263** | 1304.4564 | 1372.947632 | 1420.136963 | 0.141473 |
| 648800 | **1754.793** | 2505.1694 | 2167.084717 | 2164.018311 | 0.189104 |
| 739630 | **3801.0945** | 4393.0825 | 4291.587891 | 4325.070313 | 0.121149 |
| 761890 | 1032.8406 | 1291.7933 | 1046.670288 | **1021.805847** | 0 |
| 814380 | **1362.8702** | 1614.8129 | 1658.933594 | 1566.417725 | 0.129945 |
| 892970 | 3217.5664 | **2532.8167** | 3313.418701 | 3676.412598 | 0.311063 |
| 960090 | **1899.8359** | 2076.6377 | 2291.049316 | 2292.431396 | 0.171257 |
| 1085660 | **16838.436** | 18822.541 | 18422.99219 | 18631.20313 | 0.096224 |
| 1091500 | **13698.44** | 14906.672 | 15611.44824 | 16007.56543 | 0.144252 |
| 1172470 | **33071.402** | 39898.113 | 37495.97266 | 37135.38672 | 0.109437 |
| 1172620 | **2595.5396** | 3012.9434 | 2990.058838 | 3043.287109 | 0.147126 |
| 1222670 | 3100.3125 | **2550.8154** | 3211.132813 | 3179.268799 | 0.197672 |
| 1238810 | **1900.4874** | 1945.948 | 2186.150391 | 2113.537598 | 0.100803 |
| 1238840 | **1193.3369** | 1223.7898 | 1373.025757 | 1397.660156 | 0.14619 |
| 1293830 | 1417.9685 | 1575.0452 | **1349.839478** | 1393.606567 | 0.031406 |
| 1326470 | **1735.4362** | 1926.2095 | 2924.927979 | 2730.827881 | 0.364502 |
| 1361210 | **4610.036** | 4732.349 | 9080.219727 | 6853.158203 | 0.327312 |
| 1454400 | 809.03754 | **769.7717** | 941.6995239 | 1514.508057 | 0.491735 |
| 1623660 | **576.48615** | 741.7087 | 850.6308594 | 978.7790527 | 0.411015 |
| 1665460 | 1282.9064 | **1174.2739** | 1324.958374 | 1345.970093 | 0.127563 |
| 1677740 | 1610.0262 | 1459.8435 | **1413.402588** | 1450.460693 | 0.025549 |
| 1811260 | **4966.4424** | 9192.259 | 8108.043457 | 7732.84668 | 0.357747 |
| 1868140 | 1495.137 | **1424.7719** | 12159.14551 | 9878.760742 | 0.855774 |
| 1919590 | **1274.8295** | 2378.603 | 6667.467773 | 1967.391602 | 0.35202 |
| 1938090 | **10918.149** | 10952.329 | 14469.45508 | 14213.95605 | 0.231871 |
| 1948980 | **534.38995** | 725.96063 | 1041.738037 | 1054.809937 | 0.493378 |
| Best_count | 53 | 17 | 9 | 10 | 0.126324 |

Table 10: The mean and std of FIATS on the three dataset in metrics of MSE.

| Datasets | FIATS | FITS |
|---|---|---|
| Toy | 0.027±0.001 | 0.883±0.000 |
| Electricity | 0.193±0.004 | 0.203±0.001 |
| Weather-Medium | 0.281±0.008 | 0.430±0.011 |

## M    Error Bar & Critical Difference Diagram

We run the experiments on Toy and Electricity for five times with different randomly chosen random seeds. And Weather-Medium for three times because of the large amount of data can result in very long training time on our devices. We report the mean and standard deviation as follows with comparison with FITS, the most stable model.

As Tab. 10 indicate, FIATS shows stable performance across the benchmark. Even with extreme condition, it still maintains superior performance. It worth note that, we thought the relative large variance on weather dataset is caused by the different combination of the text description. But the FITS also shows large variance on this dataset which indicate it is hard to converge on this dataset.

We generate the critical difference plot on our result of four datasets (toy, Electricity, Weather-Medium, Weather-Large) with the default alpha as 0.05 as shown in Fig. 12. FIATS's placement at the top of the critical difference plot, without intersecting with other lines, demonstrates its consistent and superior performance in terms of MSE compared to the other models. It indicates that with the help of external textual information, FIATS can handle complicated datasets.

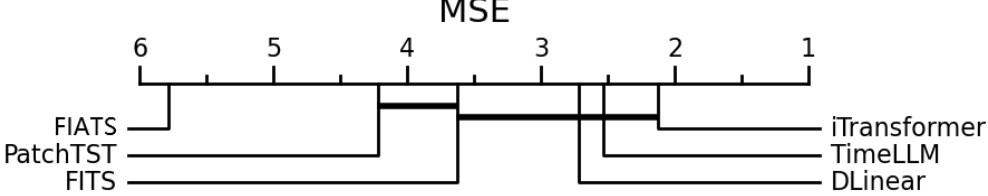

Figure 12: The Critical Difference Plot on the FIATS and other baselines with alpha=0.05.

## N    Experiment on Time-MMD

Initially, we planned to include the Time-MMD dataset as a real-world scenario to benchmark our method. However, upon evaluation, we found that this dataset suffers from significant flaws and poor organization. As a result, we have decided to include it in the appendix as supplementary material and highlight some of its issues.

### N.1    Results on Time-MMD

Despite the dataset's limitations, our FIATS demonstrates state-of-the-art performance on the Time-MMD dataset, as shown in Tab. 11. In several subsets, FIATS achieves a performance improvement exceeding 50%, showcasing its robustness and effectiveness even when applied to a flawed dataset.

### N.2    Poor Data Quality of Time-MMD

**Unbalanced Dataset.** Many subsets of the Time-MMD dataset span extended periods to collect as many valid numerical data points as possible, with some dating back to the 1980s. However, textual information from earlier periods is largely absent, resulting in a lack of corresponding records for these time intervals. In contrast, recent years have seen a surge in text articles available online. This imbalance creates challenges for the model: during training, it cannot effectively learn correlations between text and time series due to sparse or missing text data, while during inference, the model is inundated with abundant textual information. As a result, the dataset becomes inherently unbalanced.

**Meaningless Placeholder Text** Moreover, some entries in the dataset include placeholder text generated by large language models, indicating their inability to produce meaningful output due to insufficient input data. For instance, in the "Agriculture_search.csv" file, over 80% of the entries consist of statements like: "Since there is no relevant information, I am unable to provide any objective

Table 11: Results on TimeMMD.

| Agriculture | | | Climate | | |
| --- | --- | --- | --- | --- | --- |
| FIATS | Time-MMD-multi-AVG | Time-MMD-uni-AVG | FIATS | Time-MMD-multi-AVG | Time-MMD-uni-AVG |
| **0.07** | 0.09 | 0.17 | **0.24** | 1.02 | 1.24 |
| **0.11** | 0.12 | 0.19 | **0.34** | 1.02 | 1.24 |
| **0.16** | 0.15 | 0.24 | **0.43** | 1.03 | 1.24 |
| **0.17** | 0.18 | 0.29 | **0.51** | 1.03 | 1.24 |
| Economy | | | Traffic | | |
| FIATS | Time-MMD-multi-AVG | Time-MMD-uni-AVG | FIATS | Time-MMD-multi-AVG | Time-MMD-uni-AVG |
| 0.14 | **0.10** | 0.35 | **0.14** | 0.188 | 0.24 |
| 0.16 | **0.11** | 0.4 | **0.15** | 0.19 | 0.25 |
| 0.17 | **0.14** | 0.41 | **0.17** | 0.19 | 0.24 |
| 0.2 | **0.14** | 0.37 | **0.18** | 0.24 | 0.29 |
| Socialgood | | | Security | | |
| FIATS | Time-MMD-multi-AVG | Time-MMD-uni-AVG | FIATS | Time-MMD-multi-AVG | Time-MMD-uni-AVG |
| **0.66** | 0.82 | 0.87 | **68.51** | 112.76 | 118.49 |
| **0.75** | 0.93 | 0.99 | **85.81** | 115.33 | 119.09 |
| **0.78** | 1.01 | 1.07 | **89.26** | 117.19 | 121.08 |
| **0.86** | 1.05 | 1.13 | **92.89** | 118.03 | 123 |
| Energy | | | Health | | |
| FIATS | Time-MMD-multi-AVG | Time-MMD-uni-AVG | FIATS | Time-MMD-multi-AVG | Time-MMD-uni-AVG |
| **0.11** | 0.14 | 0.16 | 1.38 | **1.12** | 1.55 |
| **0.21** | 0.24 | 0.27 | 1.82 | **1.4** | 1.88 |
| **0.3** | 0.32 | 0.35 | 2.01 | **1.48** | 1.91 |
| **0.36** | 0.44 | 0.46 | 3.04 | **1.53** | 1.97 |
| Environment | | | | | |
| FIATS | Time-MMD-multi-AVG | Time-MMD-uni-AVG | | | |
| **0.32** | 0.32 | 0.35 | | | |
| **0.34** | 0.35 | 0.38 | | | |
| **0.35** | 0.37 | 0.47 | | | |
| **0.37** | 0.41 | 0.4 | | | |

facts, insights, analysis, or predictions about the United States broiler market. This search result is not relevant to United States Retail Broiler or Retail Chicken. It appears to be an advertisement for a perfume and has no connection to the topic."

Similarly, in the "Economy_search.csv", other entries state: "After reviewing the search results, I found that most of the information is not relevant to making predictions about the Economy. However, I was able to extract some useful information, which I have summarized below: NA." While these entries appear to be valid text data, they offer no meaningful or actionable information, further compounding the issue of data imbalance.

These meaningless information are all over the whole dataset, making the text validity of this dataset doubtful.

**Information Leakage.** Another significant issue with the dataset is information leakage. The dataset creators used large language models to process reports or search results and generate "fact" and "prediction" entries. However, in some cases, the reports or search results directly contain the actual values to be predicted, leading to severe information leakage.

For example, in the "Agriculture_report.csv" file, the "fact" entry for the date 2019-02-04 states: "The National Composite Weighted Average for 1/31/19 is 92.04 compared to 94.22 a week earlier, and 91.66 a year ago." This directly provides the target value to be predicted. Such instances of information leakage are pervasive throughout the dataset and significantly compromise the reliability of the results derived from it.

## O IATSF BENCHMARK DATASETS

We designed the IATSF benchmark to include four datasets of varying complexity, each tailored to evaluate specific aspects of model performance. Together, these datasets form a progression from simple, interpretable scenarios to challenging, real-world applications, providing a comprehensive evaluation framework for text-guided time series forecasting models.

*1. Toy Dataset* The Toy dataset is intentionally designed with simple and straightforward patterns, making it easy to analyze and interpret. However, the dataset includes sudden changes in patterns that are impossible to predict without text guidance. This ensures the model's ability to adhere to

textual cues is effectively tested in a controlled environment. It serves as a foundation for validating whether the model can extract and use textual guidance to forecast time series.

*2. Electricity Dataset* The Electricity dataset introduces real-world data with common textual features like day of the week or public holidays. While the textual information is relatively simple, it tests the model's ability to utilize such structured cues for forecasting. Additionally, as a widely-used off-the-shelf dataset, it allows for easy comparison with existing methods, providing a baseline for evaluating IATSF's performance.

*3. Atmospheric Physics Dataset* The Atmospheric Physics dataset represents a semi-controlled environment designed to rigorously test the model's ability to learn causal relationships between text and time series patterns. It also evaluates the model's text-guided channel independence and generalizability. By simulating a scenario where text and time series data are strongly correlated, this dataset bridges the gap between controlled tests and more complex real-world challenges.

*4. GAUD Dataset* The GAUD dataset is a fully real-world dataset that tests IATSF in a practical industrial context. Its patterns are noisy and random, making it highly challenging. This dataset showcases IATSF's ability to perform well in realistic scenarios.

**Comprehensive Benchmark Objectives**

The IATSF benchmark is designed to address multiple objectives:

- **Interpretability and Validation:** The simpler Toy and Electricity datasets help researchers validate their models and understand their behavior in controlled environments.
- **Performance Testing in Complex Scenarios:** The Atmospheric Physics and GAUD datasets challenge the models in semi-controlled and real-world settings, ensuring they are robust and capable of handling practical applications.

This benchmark is not merely a ranking tool but a framework to help researchers analyze and improve their models' behaviors across varying levels of complexity. We see this as a starting point for the community and hope it will inspire researchers to contribute additional datasets, further expanding and enriching the IATSF benchmark for future advancements in this field.

O.1  METADATA FOR DATASETS

We show the metadata for IATSF Datasets in Table 12.

Table 12: Datasets Metadata

| Dataset | Length | Time span | TS Sampling Rate | # of Channels | # of Dynamic News each step | Textual update rate | Notes |
|---|---|---|---|---|---|---|---|
| Toy | 300,000 | N/A | N/A | 1 | 1∼3 | Every Step | Sinusoidal wave with a single channel |
| Electrical Utility | 26,304 | 2011-01-01 to 2015-12-31 | 1 hour | 321 | 1∼3 | Daily | Just the Electricity Dataset |
| Atmospheric Physics | 525,600 | 2014-01-01 to 2023-12-31 | 10 minutes | 21 | 7 | Every 6 hours | Weather data with 21 channels. Three set of textual cues for combination. |
| GAUD | Varies | 2005 to 2024 | 1 Day | 1 (each game) | Varies | Varies | Each game has historical data from its prelaunch to 2024. |

O.2  TOY DATASET DETAILS

We directly generate this dataset with sinusoidal wave that randomly changes frequency. Before each changing point, we add 10 captions as 'Channel 1 will change to frequency x in y timesteps.' After each changing point, we add 5 captions as 'Channel 1 will keep steady with frequency of x.' In other timesteps, we caption it as 'The waveform will go steady.'

We will publish this dataset with CC BY-NC-SA 4.0 licence.

O.3  ELECTRICITY-CAPTION DETAILS

We caption the day of week with the given time stamp. But we somehow find the original time stamp is incorrect. Instead of the year of 2016, it should be collected in year 2012. Without knowing the

Table 13: Example caption of the Atmospheric Physics dataset.

| Topic | Example |
|---|---|
| Month & Time of the Day | It's the early morning of a day in January. |
| Overall Weather | The current weather is clear. |
| Weather Trend in next 6h | The weather is expected to remain clear. |
| Temperature Trend in next 6h | The temperature is showing a mild drop. |
| Wind Speed & Direction | There is Light Breeze from NNW. |
| Atmosphere Pressure Level | The atmospheric shows Average Pressure. |
| Humidity Level | The air is very humid. |

exact location of this building, we cannot identify the specific public holiday. We then uses channel 319, which shows obvious patterns of workday and holiday as indicator, when the average value lower than a specific value, we caption it with public holiday.

We will publish this dataset with CC BY-NC-SA 4.0 licence.

### O.4 ATMOSPHERIC PHYSICS DETAILS

#### O.4.1 DATA SOURCE

In creating a IATSF dataset, it is advisable to avoid directly generating the description out of the forecasting horizon time series pattern as news messages, as this could lead to information leakage. News messages should instead contain relevant, known information from other sources. Thus, we get the weather time series data from: `https://www.bgc-jena.mpg.de/wetter/` and weather report from `https://www.timeanddate.com/weather/germany/jena/ historic`. We will publish this dataset with CC BY-NC-SA 4.0 license since the data source forbids commercial use.

#### O.4.2 MOTIVATION

The Atmospheric Physics dataset is designed as a semi-controlled environment to rigorously test the model's ability to learn causal relationships between text and time series, as well as its text-guided channel independence and generalizability.

Such scenarios are commonly encountered in industrial applications, where correlated text and time series data often coexist. However, obtaining and releasing industrial datasets is challenging due to intellectual property restrictions. To address this, we chose the weather system—a widely available, well-understood, and publicly accessible domain—to simulate these scenarios.

As an off-the-shelf IATSF dataset, the Atmospheric Physics dataset provides a benchmark for evaluating the model's capacity to learn causal relationships between text and time series patterns, offering a practical and accessible alternative for research and experimentation.

#### O.4.3 STATISTICAL DETAIL OF THE TIME SERIES

For better understanding of the statistical distribution of Weather dataset, we plot the histogram of all 21 channels in Fig. 13.

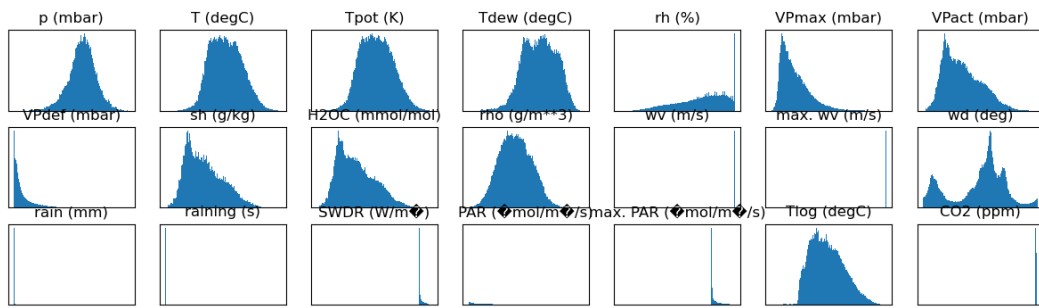

Figure 13: Histogram of all 21 channels. It shows all the channels have unique value distribution, making a model hard to generalize on all channels without knowing related information.

### O.4.4 Use of Large Language Models for Preprocessing the Weather Dataset

We would like to clarify that the use of Large Language Models (LLMs) in this work is strictly limited to the preprocessing and creation of the Atmospheric Physics dataset. LLMs are not part of our model or method, nor do they contribute to the training or inference process of FIATS. The Atmospheric Physics dataset is intended to serve as an off-the-shelf, text-time synchronized benchmark dataset with raw text and pre-embedded text embeddings as optional inputs.

The primary reason for using LLMs to preprocess this dataset is to generate diverse and correlated textual descriptions, ensuring a richer corpus for training and evaluation. By incorporating varied expressions, we enable the model to generalize to different textual forms while aligning the semantic meaning of text with time series patterns. For instance, the descriptions "The morning will be sunny, but clouds will increase in the afternoon with a chance of light rain" and "The day starts with clear skies, gradually turning cloudy with some rain in the afternoon" carry the same semantic information but differ in expression. This diversity enhances the robustness of the benchmark and validates the model's generalization capabilities.

Additionally, the raw data source for this dataset often includes general weather reports in text form, accompanied by coarse numerical updates every six hours. While numerical values such as {High_Temp: 25, Low_Temp: 20, Temp_Trend: slightly increasing, Wind_Speed: 5, Wind_Direction: East} are available, they lack the precision required for reliable exogenous variables. Moreover, the raw text contains rich semantic details—such as qualitative weather descriptions—that cannot be effectively captured using numerical values or one-hot encoding. Using text embeddings allows the model to leverage both semantic and numerical information more effectively.

In summary, the LLM preprocessing step is solely for dataset preparation and corpus diversity, ensuring that the Atmospheric Physics dataset is suitable for evaluating text-guided time series forecasting models. Our method does not rely on any LLM capabilities, and the inclusion of LLM-generated text is not a necessary step for IATSF or any similar model. We will include raw data samples in the final paper to provide greater clarity and avoid any misunderstandings.

### O.4.5 Channel Details

The meaning of each channel are as follows. The original weather dataset only contains the abbreviation for each channel, to further enrich the semantic for accurate information, we add a line of explanation after it as the channel description.

- p (mbar): Atmospheric pressure measured in millibars. It indicates the weight of the air above the point of measurement.

- T (degC): Temperature at the point of observation, measured in degrees Celsius.

- Tpot (K): Potential temperature, given in Kelvin. This is the temperature that a parcel of air would have if it were brought adiabatically to a standard reference pressure, often used to compare temperatures at different pressures in a thermodynamically consistent way.

- Tdew (degC): Dew point temperature in degrees Celsius. It's the temperature to which air must be cooled, at constant pressure and water vapor content, for saturation to occur. A lower dew point means dryer air.

- rh (%): Relative humidity, expressed as a percentage. It measures the amount of moisture in the air relative to the maximum amount of moisture the air can hold at that temperature.

- VPmax (mbar): Maximum vapor pressure, in millibars. It represents the maximum amount of moisture that the air can hold at a given temperature.

- VPact (mbar): Actual vapor pressure, in millibars. It's the current amount of water vapor present in the air.

- VPdef (mbar): Vapor pressure deficit, in millibars. The difference between the maximum vapor pressure and the actual vapor pressure; it indicates how much more moisture the air can hold before saturation.

- sh (g/kg): Specific humidity, the mass of water vapor in a given mass of air, including the water vapor. It's measured in grams of water vapor per kilogram of air.

- H2OC (mmol/mol): Water vapor concentration, expressed in millimoles of water per mole of air. It's another way to quantify the amount of moisture in the air.

- rho (g/m³): Air density, measured in grams per cubic meter. It indicates the mass of air in a given volume and varies with temperature, pressure, and moisture content.

- wv (m/s): Wind velocity, the speed of the wind measured in meters per second.

- max. wv (m/s): Maximum wind velocity observed in the given time period, measured in meters per second.

- wd (deg): Wind direction, in degrees from true north. This indicates the direction from which the wind is coming.

- rain (mm): Rainfall amount, measured in millimeters. It indicates how much rain has fallen during the observation period.

- raining (s): Duration of rainfall, measured in seconds. It specifies how long it has rained during the observation period.

- SWDR (W/m²): Shortwave Downward Radiation, the amount of solar radiation reaching the ground, measured in watts per square meter.

- PAR (umol/m$\hat{2}$/s): Photosynthetically Active Radiation, the amount of light available for photosynthesis, measured in micromoles of photons per square meter per second.

- max. PAR (umol/m$\hat{2}$/s): Maximum Photosynthetically Active Radiation observed in the given time period, indicating the peak light availability for photosynthesis.

- Tlog (degC): Likely a logged temperature measurement in degrees Celsius. It could be a specific type of temperature measurement or recording method used in the dataset.

- CO2 (ppm): Carbon dioxide concentration in the air, measured in parts per million. It's a key greenhouse gas and indicator of air quality.

### O.4.6 VISUALIZATION OF THE TEST SAMPLE

We show a segment of test sample along with the dynamic news timeline in Fig. 14. The news messages are sparse and vague and not directly correlated to some of the channels. These text are passed to the model as text embeddings and aligned with time series on time domain. Thus, the model can extract causal relationship to guide each channel to perform accurate prediction even though they have distinguished distribution.

The following sections give detailed performance and visualization across all the channels.

### O.5 GAUD DETAILS

GAUD datasets contains 89 subdataset. Each subdataset correspond to the active user time series of one specific game along with text information. Each subdataset contains one basic information as the channel description includes game title, genera and developer also with the update log includes release date, update type, update title, and article body. We will release the pre embeddings of these text information to avoid violating intellectual property constrains.

## P IMPLEMENTATION DETAILS AND HYPER-PARAMETERS

We train our model on single NVIDIA A800 GPU.

For electricity dataset, we directly report the result from the original paper. For weather dataset, we uses the exact set of hyper-parameter for the original weather datasets provided by each baseline model.

In most of the experiments, we simply use a patch length of 6 and stride of 3. For Toy dataset, we use patch length of 16 and stride of 8.

We follow the previous works, split all the dataset by 7:1:2 for training, validation and testing.

Except the performance on the Atmospheric Physics Dataset, all other experiments are ran on the MiniLM Embedding. We selected MiniLM as the embedding model because it achieves results comparable to OpenAI embeddings while producing smaller embeddings (384 dimensions for

["It's the morning of a day in April.",
'The current weather is sunny. ',
'The weather will transition from clear to sunny,
with passing clouds observed later.',
'The temperature is gradually increasing.',
'There is Light Breeze from SSW.',
'The atmospheric pressure is high.',
'The air is going from very humid to humid.']

["It's the evening of a day in April. ",
'The current weather is clear. ',
'The weather is expected to remain clear due to the
absence of detailed forecasts. ',
'The temperature is within a moderate range, with no
significant changes reported. ',
'There is Light Breeze from N. ',
'The atmospheric shows Average Pressure. ',
'The humidity is somewhat humid.']

["It's the morning of a day in April.",
'The current weather is rain showers with
some sunny spells. ',
'The weather is transitioning from clear to
rain showers, becoming partly sunny towards
the end. ',
'The temperature remains steady. ',
'There is a Light Breeze from NNW. ',
'The atmospheric pressure is average. ',
'The air is extremely humid.']

["It's the evening of a day in April. ",
'The current weather is partly cloudy.',
'The weather is expected to remain unchanged,
without detailed forecast available.',
'The temperature range suggests a mild evening ahead.
',
'There is Light Breeze from W.',
'The atmospheric shows High Pressure.',
'The humidity is very high.']

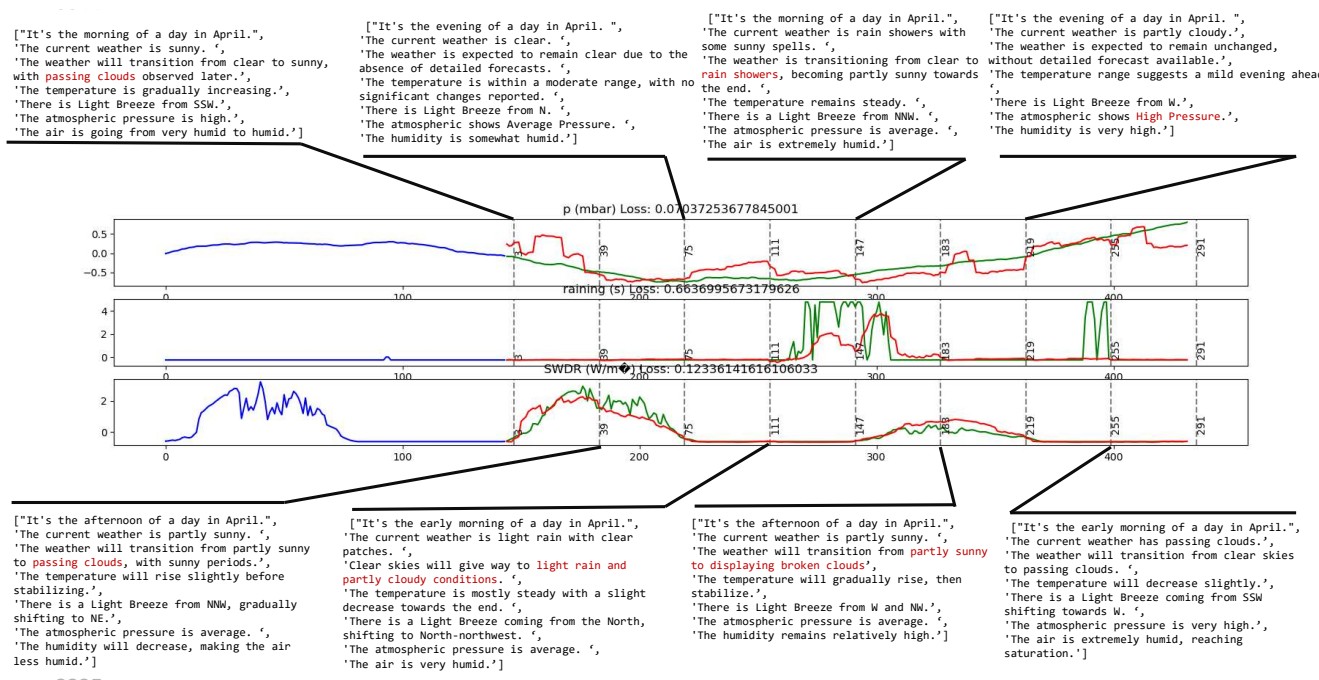

["It's the afternoon of a day in April.",
'The current weather is partly sunny. ',
'The weather will transition from partly sunny
to passing clouds, with sunny periods.',
'The temperature will rise slightly before
stabilizing.',
'There is a Light Breeze from NNW, gradually
shifting to NE.',
'The atmospheric pressure is average. ',
'The humidity will decrease, making the air
less humid.']

["It's the early morning of a day in April.",
'The current weather is light rain with clear
patches. ',
'Clear skies will give way to light rain and
partly cloudy conditions. ',
'The temperature is mostly steady with a slight
decrease towards the end. ',
'There is a Light Breeze coming from the North,
shifting to North-northwest. ',
'The atmospheric pressure is average. ',
'The air is very humid.']

["It's the afternoon of a day in April.",
'The current weather is partly sunny. ',
'The weather will transition from partly sunny
to displaying broken clouds',
'The temperature will gradually rise, then
stabilize.',
'There is Light Breeze from W and NW.',
'The atmospheric pressure is average. ',
'The humidity remains relatively high.']

["It's the early morning of a day in April.",
'The current weather has passing clouds.',
'The weather will transition from clear skies
to passing clouds. ',
'The temperature will decrease slightly.',
'There is a Light Breeze coming from SSW
shifting towards W. ',
'The atmospheric pressure is very high.',
'The air is extremely humid, reaching
saturation.']

Figure 14: An visualization of test sample with all the corresponding dynamic news. Atmospheric Physics dataset have dynamic weather report update every 6 hours. As we demonstrate a case of predicting 48 hours. Note that the embedding of these sentences are fed to the FIATS along with the look-back window time series as input. We highlight some of the words that may make impact on the forecasting result.

MiniLM versus 512 for OpenAI). This reduced embedding size speeds up training, particularly for ablation studies, making it more practical for our experiments.

Further detailed hyperparameter settings are provided in the training scripts in our codebase. We did not perform comprehensive hyper-parameter tuning because of the constraint of compute power. Thus, we may report a sub-optimal result of FIATS.

