# OpenReview forum: "Influence-Aware Forecasting: Breaking the Self-Stimulation Barrier in Time Series"
_ICLR.cc/2026/Conference — Submitted to ICLR 2026_

### Official Review · Reviewer_BWDJ · 2025-10-25

**Soundness:** 2
**Presentation:** 3
**Contribution:** 2
**Rating:** 4
**Confidence:** 3

**Summary:**

Paper attributes the performance bottleneck of TSF to the “self-excitation” assumption and introduces the IATSF paradigm along with the lightweight model FIATS. By incorporating external influences such as text as conditional inputs, it seeks to lower the theoretical lower bound of prediction error. On the theoretical side, it establishes a lower bound on error when external factors are ignored and proves that “partial influence can also tighten the bound.” On the engineering side, it builds a “time-synchronized, leakage-free” benchmark and implements channel sensitivity and conditional decoding via CASM/CAPS. Experiments on synthetic, meteorological, traffic, and market datasets report improvements over common baselines.

**Strengths:**

Paper explicitly identify how the “self-stimulation” assumption induces averaging effects and imposes an intuitive lower bound on error, and paper provide a precise expression for this bound.

The engineering design and lightweight fusion modules (CASM/CAPS) achieve strong results on selected datasets and exhibit a degree of interpretability.

**Weaknesses:**

The selection of exogenous inputs lacks formal criteria; choices are overly subjective and do not employ statistical or causal tests to determine whether an influence is relevant or actionable. There is no assessment of ex ante testability or ex post consistency, making the approach susceptible to selection bias.

The scope is narrow, being effective primarily for datasets that are directly and strongly driven by external factors; no compelling evidence is provided for weakly correlated settings or systems with complex feedback.

The chosen exogenous inputs are uniformly idealized, without accounting for the negative impacts of irrelevant external inputs.

Even on the most favorable dataset, the MSE is only 0.003 lower than the second-best result, and no standard deviations are reported, casting doubt on the stability of the findings.

**Questions:**

What is FIATS’s worst-case behavior when external influences are weakly correlated or irrelevant? Can it at least match the performance of a model without external influences?

Is there a principled framework—statistical, causal, or otherwise—that offers theoretical guidance for selecting which external influences to include?

Can the theory be extended to accommodate weakly correlated external influences?

---

### Official Review · Reviewer_bRBF · 2025-10-31

**Soundness:** 1
**Presentation:** 2
**Contribution:** 2
**Rating:** 2
**Confidence:** 5

**Summary:**

The manuscript presents a very interesting study focused on improving time series prediction.
The primary motivation for this manuscript is that traditional approaches are based on the "self-stimulation" assumption, i.e., predictions are made solely based on previous observations.
This manuscript provides two main contributions: (i) a temporally-synced benchmark that incorporates textual influences to capture the qualitative or uncertain dynamics missed by traditional variables; (ii) FIATS, a lightweight, principled model
engineered to interpret these influences.

**Strengths:**

The main positive aspect of this proposal lies in combining LLM to add new information and improve the performance of predictors. This trend has been observed in several top-tier conferences. As discussed in the manuscript, there are important related works addressing this problem.The evaluation of contributions in time series collected from different sources and with varying complexities is also a positive aspect.

**Weaknesses:**

- The manuscript cites a very important related work: "Context is key: A benchmark for forecasting with essential textual information" (Williams et al., 2025). However, this paper should be used as a benchmark.
- Still related to this work, and others related, the manuscript says: "the approaches often lack a clear theoretical justification for how influences should be modeled". However, the theoretical justification presented by the authors has two main issues: mostly based on two propositions; and the covariance error bound does not justify the paper's hypothesis.
- Experiments only using MSE allow for monitoring just a single type of error. No causal relation was measured, for example.
- The manuscript proposes the inclusion of new information to improve predictions, but only univariate models without considering, for example, other multimodal models.
- Another important missing discussion is about the variation of horizon predictions.

**Questions:**

- Why were the most related works not evaluated as benchmarks? For example, (Williams et al., 2025). The comparison presented in the manuscript was performed only considering univariate models. Explain why multivariate and multimodal baselines were excluded, or include them to ensure a fair assessment.

- The main motivation must be more precise. The idea of combining LLM and time series is not new. Therefore, the paper should articulate concrete problems with the current approach and how to solve them.
- Still about the previous comment, the justification of "lack a clear theoretical justification" is not properly addressed by the authors.
- Considering the sentence "Proposition 3.1 demonstrates that any measurable influence information reduces forecasting uncertainty, even with incomplete influence knowledge. This motivates our key insight: textual descriptions
of influences provide viable information for uncertainty reduction, despite non-numeric formats." is not addressing the authors' motivation about the lack of a clear theoretical justification. At least, it is not enough to show general variation of the covariance error bound suggested in the manuscript.
- Please, consider using other regression metrics to better understand the prediction errors.
- Discuss how the proposal is impacted by short- and long-term predictions.
- Equations must be revisited to avoid imprecision of definitions: In Line 259, should the last term be $U_f$, considering the equation in Line 105? What's k in Line 306? Same as in Line 313? Was $B_{U_f}$ in Line 270 correctly defined as in Line 259?
- In Section 6.1, the authors mention "the limitationS of self-stimulation". Clarify them, once I've just noticed one limitation.
- All quotation marks in latex are wrong, especially the single ones.

---

### Official Review · Reviewer_7V6c · 2025-10-31

**Soundness:** 3
**Presentation:** 3
**Contribution:** 2
**Rating:** 4
**Confidence:** 3

**Summary:**

The paper reframes time-series forecasting from “self-stimulation” (history-only) to Influence-Aware TSF (IATSF), pairing future-relevant external influences with a leak-controlled benchmark and a lightweight model (FIATS) that uses CASM/CAPS for channel-aware modulation. Experiments across synthetic, physics, market, and traffic data show consistent gains, with ablations indicating the improvements come from the influence inputs and channel descriptions rather than model size.

**Strengths:**

- Clear problem framing and theory: motivates the error floor of history-only forecasting and why conditioning on influences helps.
- Time-aligned influences to avoid future information leakage.
-  FIATS with CASM/CAPS offers principled channel-aware conditioning without heavyweight models.
- Consistent empirical gains: improvements on diverse datasets and horizons, also the interpretable attention surfaces add insight.

**Weaknesses:**

- Most baselines are classic self-stimulated TSF models. It isn’t shown that they’re given equivalent exogenous inputs (e.g., weather forecasts / channel descriptions) or simple text adapters, so some gains may come from extra information rather than architecture. Please add baselines that receive the same influence features (PatchTST + forecast covariates, or text-embedding concatenation), and exogenous-aware TSF baselines. If FIATS is the only model recieving more information that other models, it is not surprising that it performs better than others
- The model adds CASM/CAPS blocks and a token-wise decoder. It would help to report parameter counts or FLOPs and compare with othe models to get a better sense of performance in terms of model size.
- IATSF assumes influences are independent of the system state and take effect instantaneously. Real systems (traffic, power demand, weather-to-physics) often have lagged and endogenous effects. The main text does not quantify sensitivity to these violations.
- Evaluation is mostly MSE. Many TSF applications also use MAE, MAPE, and SMAPE, etc.
- The paper itself shows that when trained on good text but tested on noisy/misleading text, performance degrades notably—suggesting susceptibility to imperfect inputs that are common in practice. Therefore, it would be limited to only test on synthetic noises. Would benefit from quantifying robustness on real forecast and text errors.

**Questions:**

- Are the text embeddings frozen? What did you do to bridge the domain adaptation? Given OpenAI embeddings appear strongest in Table 3, can you ensure results are reproducible with open models only?
- It would be good to provide per-channel and per-series metrics to verify that gains aren’t concentrated in a few channels.

---

### Official Review · Reviewer_GPeT · 2025-10-31

**Soundness:** 3
**Presentation:** 2
**Contribution:** 3
**Rating:** 4
**Confidence:** 3

**Summary:**

This paper proposes an "Influence-Aware Forecasting" framework aimed at breaking through the long-standing limitations of the "self-exciting" assumption in time series forecasting. From the perspective of control theory, the authors point out that traditional models overlook external influences, leading to bottlenecks in predictive performance. They introduce a new approach that integrates external intervention information (such as text, images, events, etc.). Experimental results demonstrate that this method significantly outperforms existing mainstream models on multiple real-world datasets, particularly showing greater robustness when dealing with unexpected events or external interventions.

**Strengths:**

1.Theoretical Innovation: For the first time, this work analyzes the structural bottlenecks in time series forecasting from the perspective of control theory, offering insightful perspectives.
2.Challenging Mainstream Assumptions: It breaks the "self-exciting" paradigm, opening up new directions for time series research.

**Weaknesses:**

1. The details of external influence modeling are somewhat vague: the paper does not elaborate on how to select, process, and integrate multimodal external information, which may affect reproducibility.
2. Computational costs are not sufficiently discussed: introducing external data could significantly increase training and inference costs, yet the paper does not evaluate its resource consumption.
3. The choice of comparison methods is somewhat conservative: in some experiments, the baselines used for comparison are relatively weak, and there is a lack of in-depth comparisons with the latest large models (such as TimePro, TimeMixer++, etc.).
4. The author raises the issue of self-motivation and claims to be able to solve it. However, the text prompts are also part of the currently available information, which actually includes similar information in historical data. Therefore, adding text prompts does not effectively address the problem. This motivation is flawed.

**Questions:**

see the weakness

---

### Meta-Review · Area_Chair_Uti6 · 2026-01-04

**Summary:**

This paper proposes *Influence-Aware Time Series Forecasting (IATSF)*, arguing that the performance bottleneck in time series forecasting stems from the widely adopted history-only (“self-stimulation”) assumption. It presents a control-theoretic analysis of an irreducible error bound when external influences are ignored, and introduces a leakage-free benchmark together with a lightweight influence-conditioned model (FIATS).



Although the conceptual premise is interesting, the submission suffers from fundamental flaws in both theoretical scope and empirical rigor. Reviewers unanimously questioned the validity of the experimental results, noting that the proposed method may benefit from look-ahead bias that is unavailable to baselines. Furthermore, critical experimental details are missing. As a result, the baseline comparisons appear inequitable, and the authors' claims regarding efficiency, generalization, and a resultant 'paradigm shift' are not supported by the evidence provided.

**Reviewer Concerns:**

**Theoretical limitations**: The analysis relies on idealized assumptions (independent, instantaneous, exogenous influences) and does not adequately address delayed, endogenous, or weakly correlated influences common in real-world systems, limiting the scope of the conclusions.

**Baseline fairness**: Competing methods are not provided with equivalent access to exogenous or textual information, making it unclear whether gains arise from the proposed modeling framework or simply from additional information.

**Experimental scope**: Evaluation relies primarily on MSE, with limited use of alternative metrics (e.g., MAE, SMAPE), per-channel analysis, or robustness tests under realistic noise or imperfect influence forecasts.

**Influence selection and realism**: The choice of external influences lacks a principled or causal criterion, raising concerns about selection bias, scalability, and applicability to more weakly driven systems.

**Empirical completeness**: Missing comparisons with stronger multimodal or exogenous-variable baselines, limited discussion of computational cost, and some presentation/clarity issues further weaken the empirical claims.

While the motivation is potentially impactful, the proposed problem bears a striking resemblance to existing literature without citation. Furthermore, the current theoretical justification and empirical validation are not sufficiently mature to support the claims at ICLR standards.

**Reviewer Scores:**

All scores below the acceptance bar. Major concerns regarding theoretical completeness and experimental rigor were raised, prompting a recommendation of strong rejection from one reviewer.

---

### Decision · Program_Chairs · 2026-01-26

Reject